# The homeodomain regulates stable DNA binding of prostate cancer target ONECUT2

Avradip Chatterjee [1,2], Brad Gallent [2,3], Madhusudhanarao Katiki [1,2], Chen Qian[2,3], Matthew R. Harter[1,2], Steve Silletti[4], Elizabeth A. Komives [4], Michael R. Freeman [2,3] ✉ & Ramachandran Murali [1,2] ✉

The CUT and homeodomain are ubiquitous DNA binding elements often tandemly arranged in multiple transcription factor families. However, how the CUT and homeodomain work concertedly to bind DNA remains unknown. Using ONECUT2, a driver and therapeutic target of advanced prostate cancer, we show that while the CUT initiates DNA binding, the homeodomain thermodynamically stabilizes the ONECUT2-DNA complex through allosteric modulation of CUT. We identify an arginine pair in the ONECUT family homeodomain that can adapt to DNA sequence variations. Base interactions by this ONECUT family-specific arginine pair as well as the evolutionarily conserved residues are critical for optimal DNA binding and ONECUT2 transcriptional activity in a prostate cancer model. The evolutionarily conserved base interactions additionally determine the ONECUT2-DNA binding energetics. These findings provide insights into the cooperative DNA binding by CUT-homeodomain proteins.

Transcription factors, particularly master regulators, play a major role in cell-fate and tissue specification during development[1,2]. The ONE-CUT (OC) transcription factor family, consisting of OC1, OC2 and OC3 paralogs, are essential for the development of gastrointestinal organs[3–9] as well as nervous system components, including the retina[10–12] and motor neurons[13]. OC proteins feature a conserved DNA binding module comprised of a single CUT domain and a variant homeodomain (HOX), an arrangement also seen in POU transcription factors[5,10,14–16]. The CUT-HOX combination is further found in the SATB and CUX transcription factor families that contain more than one CUT domain in addition to the HOX domain. CUT, like the structurally homologous POU-specific domain, shares a similar fold as λ and 434 phage repressor DNA binding motifs[17–20], while HOX is a widespread gene regulatory element present in nearly 30% of transcription factors in humans[21,22]. Thus, CUT and HOX represent two of the most evolutionarily conserved and ubiquitous DNA binding elements essential in development. Initial structural work on POU members[19,23,24], and subsequently OC1[25], showed that the CUT and HOX bind predominantly to the opposite strands of the same major groove of DNA in an 'overlapping' manner. In addition, an isolated CUT or HOX domain shows weaker DNA interaction compared to the intact DNA binding module comprising both the domains[26,27]. However, despite these structural and biochemical data, the mechanism of DNA binding of CUT-HOX proteins is unknown. As a result, the specific roles of the CUT and HOX domains, and their coordination, in DNA binding remain poorly understood.

Previous studies have identified OC2 as a master transcription regulator driving lethal and therapy resistant prostate cancer (PC)[28,29]. In metastatic PC, aberrant overexpression of OC2 promotes treatment resistance and transdifferentiation to neuroendocrine PC (NEPC) through repression of the androgen receptor (AR) axis, and activation of *PEG10*, a known NE driver[28], as well as other oncogenic target genes. In addition, OC2 overexpression also promotes NEPC development by regulating hypoxia signaling[29]. Furthermore, an OC2 inhibitor suppressed tumor growth and metastasis in a PC xenograft mouse model[28]. OC2 is involved in other cancer types, including breast

[1]Department of Biomedical Sciences, Research Division of Immunology, Cedars-Sinai Medical Center, Los Angeles, CA, USA. [2]Samuel Oschin Comprehensive Cancer Institute, Cedars-Sinai Medical Center, Los Angeles, CA, USA. [3]Departments of Urology and Biomedical Sciences, Cedars-Sinai Medical Center, Los Angeles, CA, USA. [4]Department of Chemistry & Biochemistry, University of California San Diego, La Jolla, CA, USA. ✉e-mail: michael.freeman@cshs.org; ramachandran.murali@csmc.edu

cancer, where it similarly acts to drive lineage plasticity and credentialed as a drug target[30]. OC2 has thus emerged as an important cancer therapeutic target, and a better molecular understanding of this transcription factor is therefore of fundamental and clinical importance.

To obtain insights into DNA binding by CUT and HOX domains, and in the context of OC2 as a key mediator of PC progression, we determined the crystal structure of the human OC2 DNA-binding module (OC2 hereafter) in complex with a physiologically relevant (*PEG10*) promoter DNA sequence. To obtain further mechanistic details, we complemented our structural analyses with thermodynamics and kinetics studies. Our integrative approach reveals a detailed mechanism of the cooperativity and interplay between the CUT and HOX domains to bind DNA. We validated our results in an in vitro metastatic PC model, demonstrating the interactions we characterized to be relevant in a disease context.

## Results

### Structure of OC2 in complex with *PEG10* promoter (*PEG10*) DNA

The structure of OC2 in complex with *PEG10* promoter (*PEG10*) DNA shows the two α-helical domains, CUT and HOX, together with the connecting linker, wrap around the DNA major groove (Fig. 1a). The CUT domain (amino acids 330-407) forms five alpha helices (α1-α5) while the HOX domain (amino acids 427-481), positioned at the C-terminal of OC2, forms three α-helices (α6-α8). The linker could not be modeled due to lack of electron density suggesting it is highly flexible and does not physically bind the DNA. The helices α3 of CUT and α8 of HOX each insert into the DNA major groove (Fig. 1b, c). Compared to HOX, CUT makes more extensive contacts with the DNA, a majority of which are DNA backbone-mediated (Fig. 1d). The OC2 bound *PEG10* DNA shows only minor changes compared to a canonical B-DNA (Supplementary Fig. 1a, b).

The α3 helix mediates the bulk of DNA contacts by the CUT domain while the loops flanking both ends of α3 and those preceding α2 and α4 helices also bind the DNA. The residues Q353, S364, T367, S369, D370 and R373 of α3 helix as well as K376 in the following loop make hydrogen bonds to the DNA backbone (Fig. 1d). Q353, I351 (in the α2 helix and preceding loop, respectively), K382 and G384 (both in α4 helix) also bind the DNA backbone. S364 and Q365 residues (binding to G5′ and A3 of *PEG10*, respectively) located towards the beginning of α3 helix, and D370 (binding to C5 of *PEG10*) in the same helix are the only residues in CUT making direct base-specific hydrogen bonds with the DNA. D370, in addition, makes a water-mediated base interaction (with C6′ of *PEG10*). On the other hand, the HOX residues N476 (binding to A7 of *PEG10*), and an arginine pair, R479 and R480 (binding to T7′ and G6 of *PEG10*, respectively), form base-specific hydrogen bonds whereas R450, T469 and N472 interact with the DNA backbone.

The OC2-*PEG10* structure shares overall similarity to that of a prior OC1-*TTR* complex structure[25] (Supplementary Fig. 1c). Comparison of the DNA-bound OC2 or OC1 to a nuclear magnetic resonance (NMR)-based DNA-free structure of OC1[31] shows that the helix α3 of CUT undergoes a major reorganization upon binding the DNA (Supplementary Fig. 1d). Structural alignment of the respective apo- and DNA-bound CUT domains showed a root mean square deviation (rmsd) of 3.3 Å. This value exceeds the average rmsd of 1.5–2.5 Å observed between the structures of the same proteins elucidated by NMR and X-ray crystallography[32], suggesting the structural differences between the apo- and DNA-bound forms are induced by the DNA and independent of the respective techniques. Importantly, in the NMR structure, the beginning of α3 helix, comprising amino acids S364 and Q365, is unstructured while the helix overall is rotated by about 57° compared to that in the DNA-bound form. With respect to HOX, the apo- and DNA-bound forms do not show much structural difference, except for the C-terminal stretch beyond helix α8 not being visible in

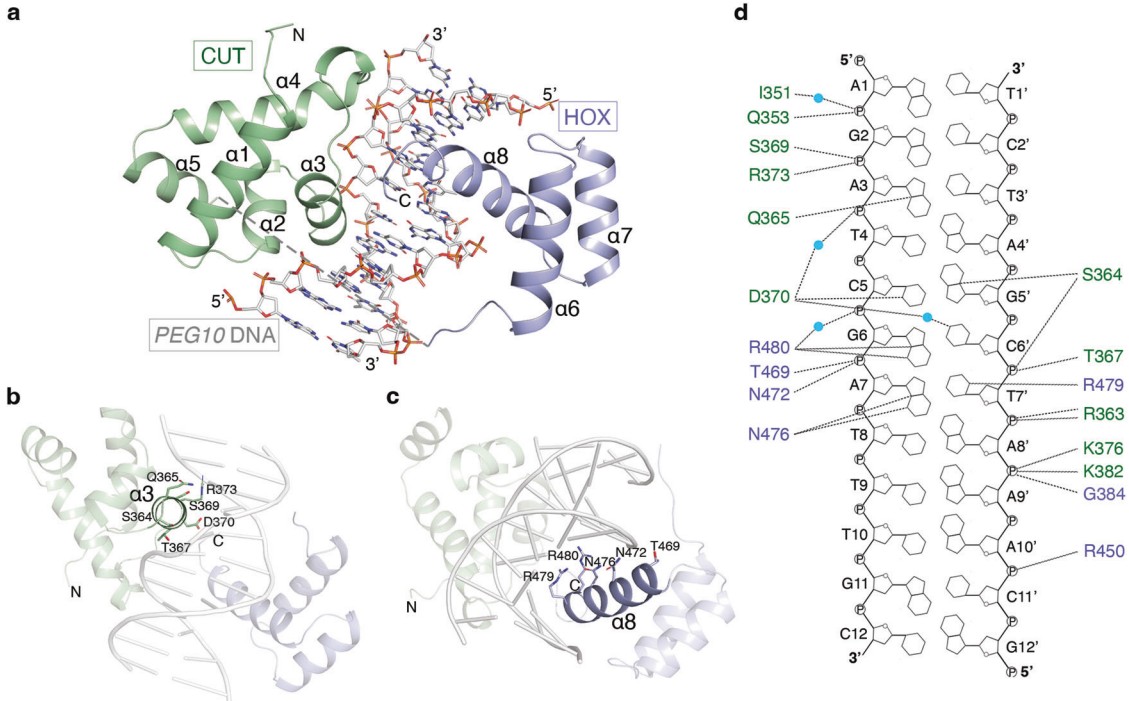

**Fig. 1 | Structure of the OC2-*PEG10* complex. a** Overall structure of the OC2-*PEG10* DNA complex. The position of CUT and HOX domains on DNA and their respective helices are labeled (α1- α8) while the unmodelled loop is depicted as a dashed line. CUT and HOX domains are shown in green and blue, respectively. **b, c** Arrangement of DNA interacting residues in α3 helix of CUT domain and α8 helix of HOX domain of OC2 are shown. **d** Schematic representation of the protein-DNA contacts in the complex. Hydrogen bonds are shown as dashed lines and water molecules are depicted as cyan spheres. DNA interacting residues of CUT and HOX domains are shown in green and blue, respectively.

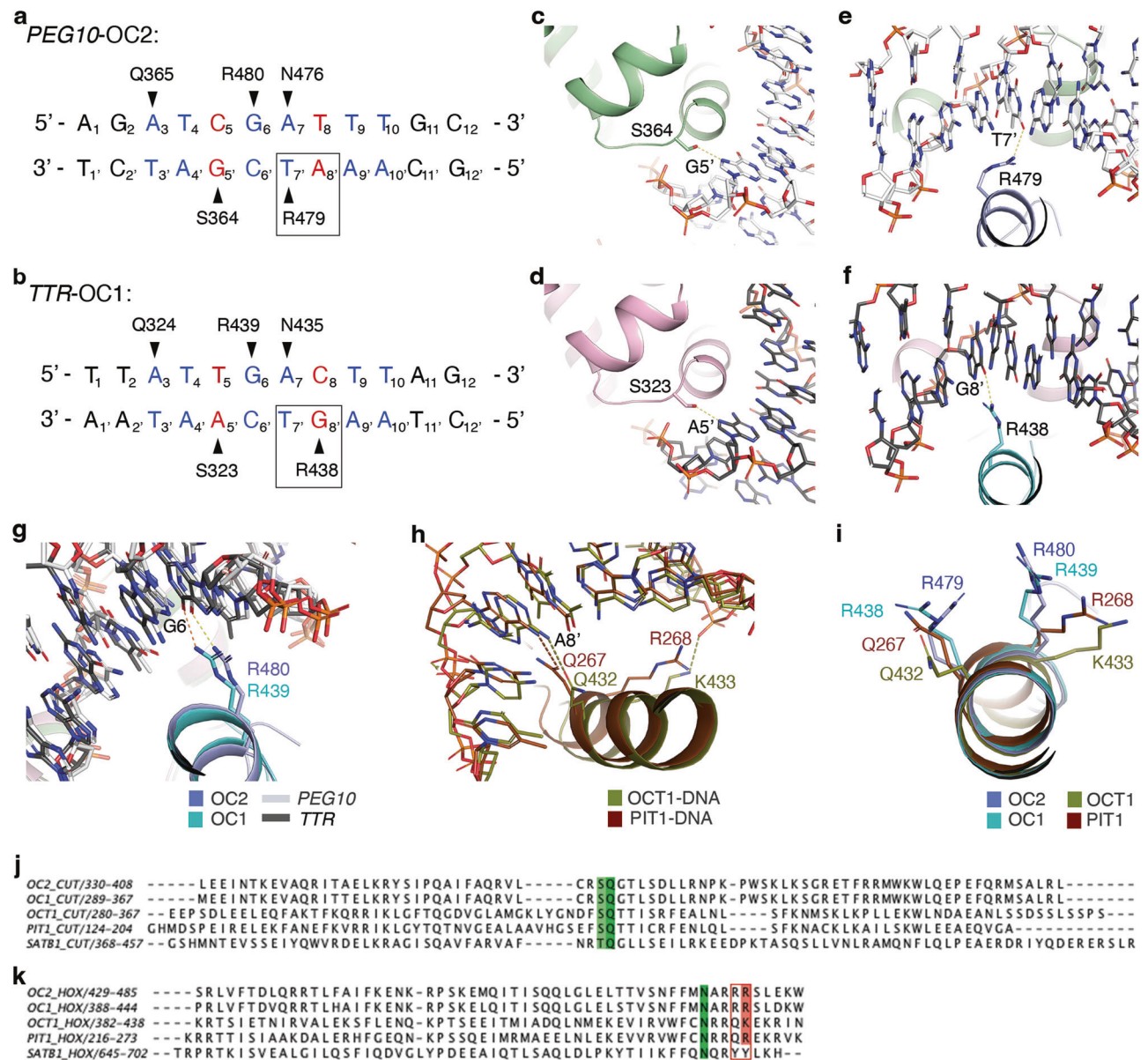

**Fig. 2 | The RR motif in the HOX domain of OC2 shows unique DNA interactions.** **a**, **b** *PEG10* and *TTR* DNA sequences and corresponding conserved base-specific interactions with OC2 and OC1, respectively. The core sequence is shown in blue except the bases at which *PEG10* and *TTR* vary, that are in red. Black triangles depict interaction sites; the respective base-interacting residues are indicated. The difference in interaction of the first arginine (OC2 R479 or OC1 R438) of RR pair is shown with a black rectangular outline. **c–f** Interaction of OC2 S364, equivalent OC1 S323, and OC2 R479 and equivalent OC1 R438, to DNA. Hydrogen bonds in panels (**c–f**) are shown as yellow dashed lines. **g** Interaction of OC2 R480 and equivalent OC1 R439 (yellow and orange dashed lines, respectively). **h** Interactions of the POU residues corresponding to the OC arginine pair [PDB 1E3O [https://doi.org/10.2210/pdb1E3O/pdb] (OCT1) and 1AU7 [https://doi.org/10.2210/pdb1AU7/pdb] (PIT1)]. Hydrogen bonds are shown in the same color as respective proteins. **i** The relative orientations of the OC arginine pair and corresponding OCT1 and PIT1 residues. **j**, **k** Structure-based sequence alignment of CUT (**j**) and HOX (**k**) domains of OC2, OC1, OCT1, PIT1 and SATB1. The amino acid ranges are indicated. The conserved serine (S364) in CUT is highlighted (in green) while the arginine pair (RR motif; R479/R480) in HOX are shown with a red border. The conserved glutamine (Q365) in CUT and asparagine (N476) in HOX are also highlighted (in green). The color of highlighted residues is based on Clustal scheme.

the apo-structure, indicating the unstructured and flexible nature of this region when not bound to DNA (Supplementary Fig. 1e).

**An arginine pair (RR motif) enables unique DNA interaction by the OC2 HOX domain**
**Differences between OC2-*PEG10* and OC1-*TTR* complex structures.** Based on our OC2-*PEG10* and previous OC1-*TTR* structures, the bulk of the interactions, including the base-specific contacts, occur through the inner eight nucleotides of bound DNA (Fig. 1d and Fig. 2a, b). A comparison of this core region of *TTR* and *PEG10* DNA shows differences mainly at two nucleotide positions – (i) at position 5, a T-A base pair in *TTR* is replaced by a C-G base pair in *PEG10* and (ii) at position 8, a C-G base pair in *TTR* is replaced by a T-A base pair in *PEG10* (Fig. 2a, b).

The adenine of (5)T-A(5') base pair in *TTR* and guanine of (5)C-G(5') base pair in *PEG10*, both being purines, form hydrogen bonds through the imidazole N7 with a conserved serine side chain oxygen present in both OC1 (S323) and OC2 (S364) (Fig. 2c, d). This serine is conserved not only within the OC family but also in CUT domains of POU members, which show relatively lower sequence similarity to the

OC family (Fig. 2j). In another related family, SATB, that contains a pair of CUT domains (and a single HOX domain) with even lower sequence conservation to the OC family than POUs, the equivalent residue is a threonine, further showing conservation at this position.

At position 8, the carbonyl oxygen of guanine of (8)C-G(8') base pair in *TTR* forms a hydrogen bond with a side chain amine of an arginine (R438) located in the helix α8 of OC1 HOX domain (Fig. 2b, f). However, in the *PEG10* sequence, the nucleotide at position 8', being an adenine, lacks the carbonyl oxygen needed to form a hydrogen bond to the equivalent side chain of arginine, R479, of OC2. Therefore, the side chain of R479 in OC2 reorients to form a hydrogen bond with the carbonyl oxygen at C4 of the preceding thymine (T7') of *PEG10* DNA (Fig. 2a, e). Notably, this arginine is part of the arginine pair (RR motif), conserved in the OC HOX domain, that mediate base-specific interactions (as mentioned in section 1; Fig. 2k). Upon comparison with other common OC-recognized promoter sequences, including *HNF-3β, HNF-4, PEPCK and PFK-2GRU*[10], we found that the base-pair at position 8 to be variable in these promoter sequences (Supplementary Fig. 2a). This suggests a general sequence variability of OC targeted gene promoters at this position and that the conserved arginine allows OC transcription factors to adapt to this variation.

**Comparison of the OC RR motif to corresponding POU residues.** Having analyzed the binding of the first arginine, as described above, we next examined the interaction of the second arginine of the pair (R480 and R439 in OC2 and OC1, respectively) to DNA. R480 in OC2 (and the equivalent R439 in OC1) forms a hydrogen bond with a guanine base (G6) (Fig. 2a, b, g). The guanine at this position is, in fact, conserved in the related promoter recognition sequences bound by the OC transcription factors mentioned above (Supplementary Fig. 2a).

The RR motif (R479/R480) although conserved within the OC family, is unique relative to POU and SATB members. In POU homeodomains, this motif consists of a glutamine followed by a lysine or arginine (QK/R) whereas in that of SATB it is made of a tyrosine pair (YY) (Fig. 2k). In POU, the glutamine (Q432 and Q267 in human OCT-1 and PIT-1 proteins) corresponding to OC2 R479, forms a hydrogen bond invariably with an adenine in the cognate promoters (Fig. 2h and Supplementary Fig. 2b). The subsequent residue in POU, which is a lysine (K433 in OCT-1) or an arginine (R268 in PIT-1), corresponding to OC2 R480, does not make base contact but can either bind to DNA backbone phosphate or remains unbound (Fig. 2h and Supplementary Fig. 2b), consequently exhibiting a difference in orientation relative to OC2 R480 (or OC1 R439) (Fig. 2i). Notably, the base at position 6 in POU specific promoter sequences is generally a cytosine, unlike the corresponding guanine (G6) in OC recognized sequences, which is bound by the second arginine of the RR motif as mentioned above (Supplementary Fig. 2b).

The above analyses indicate that the first arginine of the RR motif, through its ability to reorient, confers a degree of flexibility to OC2 and OC1 proteins to adapt to cognate base variability in the promoters of OC target genes. In addition, the second arginine, through its interaction with a conserved guanine (G6) in these promoters, provides sequence selectivity. Furthermore, these base contacts by the RR motif are unique compared to the interactions mediated by the corresponding QK/R stretch found in the POU homeodomains. Importantly, homeodomains in general, display considerable variability in the amino acids corresponding to the RR motif of OC2. For example, in yeast MATα, the first residue is an arginine whereas in *Drosophila* homeodomain proteins Engrailed (Eng) and Antennapedia (Antp), this residue is replaced by methionine and alanine, respectively. At the second position, however, there is a lysine in all three homeodomains (Supplementary Fig. 4b). Notably, apart from the arginine in MATα and methionine in Eng, which interact in a base-specific manner, the other residues mentioned above do not exhibit base-specific binding to the

DNA[33]. Taken together, the above observations lead us to propose that this arginine pair represents an important DNA base interacting motif in the HOX domain of OC2.

**The HOX domain thermodynamically stabilizes OC2 on DNA**
To understand the mechanism of OC2-DNA interaction further, we carried out thermodynamics analysis of complex formation using isothermal titration calorimetry (ITC). OC2 bound to *PEG10* DNA with a binding affinity ($K_D$) of 7 nM and an associated free energy (ΔG) of −11.2 kcal/mol. The binding is characterized by a relatively large and favorable enthalpy change (ΔH = −15.5 kcal/mol) and an unfavorable entropy (-TΔS = 4.4 kcal/mol) (Fig. 3a; Table 1 and Supplementary Fig. 3a). The favorable enthalpy change suggests stable hydrogen bonds and van der Waals interactions being formed while the unfavorable entropy signifies loss of conformational freedom during complex formation. Furthermore, the large favorable enthalpy change compensates for unfavorable entropy resulting in an enthalpically driven interaction.

We next sought to understand the roles of the individual CUT and HOX domains towards interaction with *PEG10*. For this, we expressed the CUT and HOX domains separately. The CUT domain bound *PEG10* DNA with nearly 290-fold weaker affinity showing a $K_D$ of 2030 nM and consequently a significantly lower associated ΔG (−7.8 kcal/mol). Importantly, compared to intact OC2, we observed a markedly lower enthalpy change (ΔH = −3.4 kcal/mol), and strikingly the binding showed a favorable entropy (-TΔS = −4.4 kcal/mol) (Fig. 3b; Table 1 and Supplementary Fig. 3a). These values show a smaller enthalpic and a relatively significant entropic contribution to the overall DNA binding by CUT, indicating a distinct thermodynamic pattern compared to that observed with the intact OC2. We next tested whether the OC2 HOX domain alone can bind to *PEG10* DNA but observed no binding in this case. We also did not observe any direct binding between CUT and HOX domains in the absence of DNA (data not shown), consistent with the DNA bound and unbound OC structures.

We then tested whether the presence of HOX, in addition to CUT, but as separate polypeptides, can recapitulate binding of intact OC2 to DNA. We observed a marginal improvement in DNA binding affinity ($K_D$ = 1380 nM) in the presence of HOX, suggesting that the covalent linkage between the domains provided by an intact linker is needed for the higher (7 nM) binding affinity observed with the intact OC2. Remarkably though, compared to CUT alone, we observed a marked increase in enthalpy change (ΔH = −15 kcal/mol) and an unfavorable entropy (-TΔS = 7 kcal/mol) (Fig. 3c; Table 1 and Supplementary Fig. 3a), a pattern that resembles the one observed with intact OC2. This pronounced thermodynamic shift, compared to the DNA binding of CUT alone, cannot be explained based on an additive effect caused by the HOX mediated interactions but is rather indicative of the CUT-HOX cooperativity. In addition, this thermodynamic signature of favorable enthalpy and unfavorable entropy is known to correlate with ligand-induced conformational rearrangements resulting in folding of secondary structure elements[34], for example, the DNA-induced changes in GCN4 transcription factor[35,36] and the CD4 receptor induced modulation of human immunodeficiency virus (HIV) gp120[37,38]. To investigate any underlying conformational change in OC2 upon DNA binding, we calculated the heat capacity (enthalpy change per mole per unit temperature change; ΔC°) of the interaction. A large negative ΔC° (generally ≥ −200 cal/mol/K) indicates protein folding upon interaction with the ligand[39,40]. We therefore determined the enthalpies (ΔH) of OC2-*PEG10* binding at temperatures 12, 25 and 30 °C (Table 2 and Supplementary Fig. 3b). Based on this analysis, we calculated a ΔC° of ~ −440 cal/mol/K for the interaction. Our ITC binding studies showed the CUT binding to DNA is less stable in comparison to that by OC2, so we also calculated the heat capacity for the CUT-DNA complex using the same method (Table 2 and Supplementary Fig. 3b). However, in this case, we obtained a much lower ΔC° (−74 cal/mol/K),

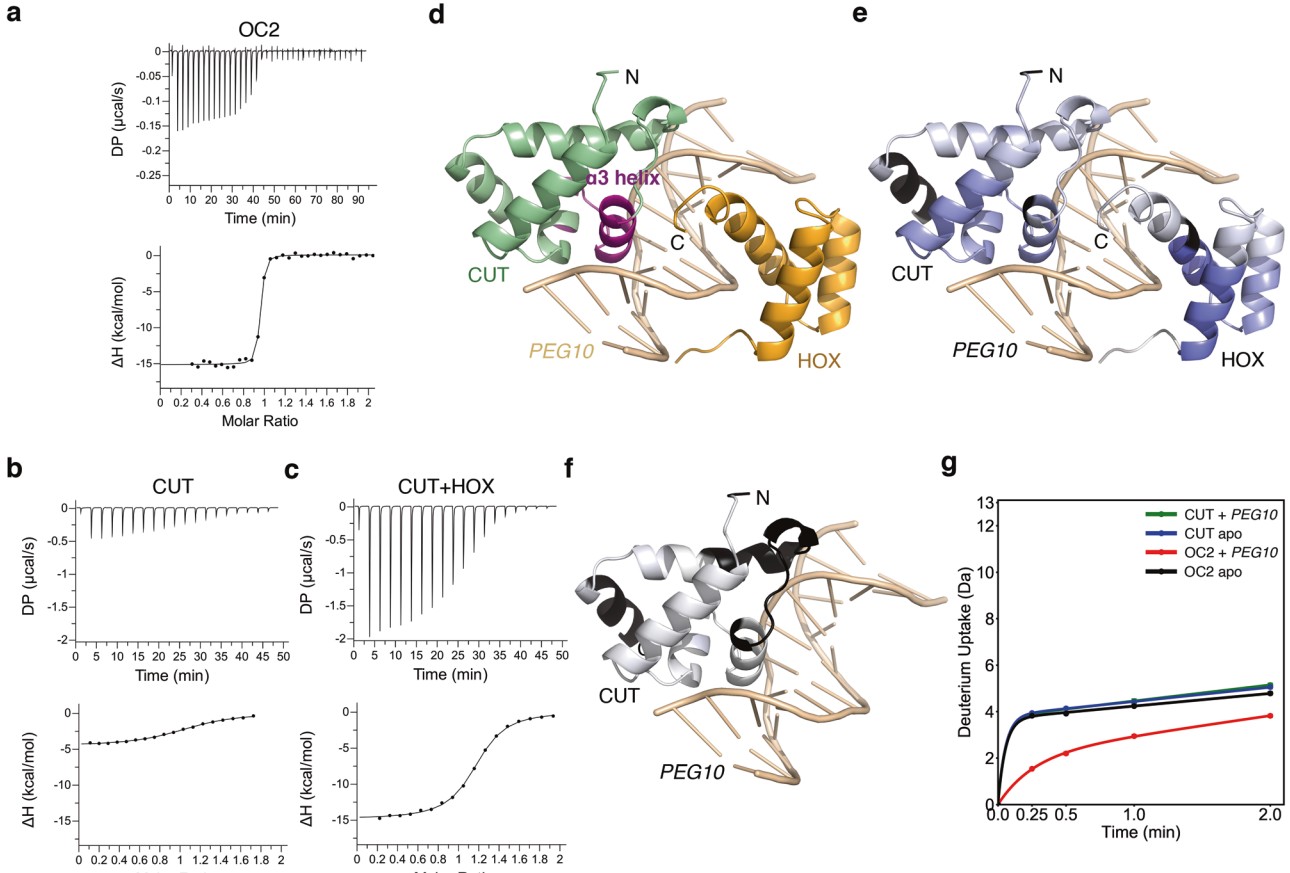

**Fig. 3 | The HOX domain drives stable association of OC2 to DNA. a–c** ITC binding analysis of intact OC2, CUT domain and CUT + HOX, to *PEG10* DNA. The raw heats (differential power, DP; top) and binding isotherms (bottom) are shown and representative of three independent experiments (*n* = 3; technical replicates). Source data are provided as a Source Data file. **d–g** Amide hydrogen-deuterium exchange mass spectrometry (HDX-MS) characterization of DNA binding by OC2. Structure of OC2 bound to *PEG10* DNA showing the helix α3 (purple) which appears to become much more structured upon *PEG10* binding (**d**). The CUT and HOX are in green and gold, respectively, while the *PEG10* is in wheat. The protein N- and C-termini are indicated. Relative deuterium uptake into OC2 with and without bound *PEG10* is rendered on the structure using a blue-white-red gradient scale (**e**); regions of the protein not covered by the HDX-MS experiment are in black. No regions of increased uptake were observed but many parts of the protein

experienced decreased (blue) deuterium uptake upon *PEG10* binding with the maximum decrease being 0.28. Relative deuterium uptake into the CUT domain alone with and without *PEG10* is plotted on the structure according to the same scale and color scheme as in panel **e** (**f**). There were no significant differences in deuterium uptake upon *PEG10* binding of the CUT domain alone (consequently, mostly white except the black regions that showed no coverage). Note that the CUT(-DNA) structure is extracted from the OC2-*PEG10* structure for demonstration only. Deuterium uptake plot for the helix α3 containing peptide (residues 358-371) in the four different conditions as indicated (**g**). The *y*-axis corresponds to the total number of amides in the peptide. The HDX-MS data shown is representative of three independent experiments (*n* = 3; technical replicates); *P*-values based on One-way ANOVA as provided by the software were 0.001 for OC2-*PEG10* vs apo-OC2; 0.9 for CUT-*PEG10* vs apo-CUT.

**Table 1 | ITC analysis of OC2, CUT and HOX binding to the *PEG10* DNA**

| Interaction | $K_D$ (nM) | ΔG (kcal/mol) | ΔH (kcal/mol) | -TΔS (kcal/mol) | N (sites) |
|---|---|---|---|---|---|
| OC2 - *PEG10* | 7 ± 2 | −11.2 ± 0.2 | −15.5 ± 0.3 | 4.4 ± 0.2 | 1.0 ± 0.1 |
| CUT - *PEG10* | 2030 ± 262 | −7.8 ± 0.1 | −3.4 ± 0.2 | −4.4 ± 0.3 | 1.2 ± 0.1 |
| (CUT + HOX) - *PEG10* | 1380 ± 279 | −8.0 ± 0.1 | −15.0 ± 1.1 | 7.0 ± 1.0 | 1.2 ± 0.2 |

that is generally not associated with structural changes upon DNA binding[40]. These observations imply that indeed OC2, unlike isolated CUT, undergoes DNA-dependent folding.

Comparison between the DNA bound and unbound OC structures indicates a rearrangement in the CUT domain, especially the α3 helix region, upon DNA binding (Supplementary Fig. 1d). To further confirm these conformational changes, based on the ITC experiments as well as structural analyses, we employed hydrogen-deuterium exchange mass spectrometry (HDX-MS) of OC2 and the isolated CUT domain, alone (apo-) or as respective *PEG10* DNA bound complex. A protection of the α3 helix was observed only in DNA bound OC2 but not in apo-OC2,

apo-CUT and DNA bound CUT (Fig. 3d–g and Supplementary Fig. 3c, d). These results indicate that the α3 helix in intact OC2, unlike in CUT, indeed undergoes structural rearrangement upon DNA binding, which allows OC2 to bind DNA stably with higher affinity.

In summary, our thermodynamic observations followed by HDX-MS analysis suggest structural rearrangements in OC2 upon binding to *PEG10* DNA and imply these changes to be dependent on HOX. Furthermore, considering the lack of physical interaction between CUT and HOX, the above observations indicate the rearrangements in CUT being induced allosterically by HOX, through the DNA, leading to a stable CUT-HOX-DNA ternary complex.

### S364/Q365 and N476 mediated base interactions are necessary for correct DNA-bound conformation of OC2

An understanding of the role of base-specific interactions in overall DNA binding by OC, and CUT-HOX transcription factors in general, is unclear. Therefore, based on the above thermodynamics insights, we sought to understand the contribution of the conserved base-specific interactions towards the DNA binding of OC2. We introduced relevant alanine mutations in both CUT and HOX domains, in the context of the intact OC2 DNA binding module. As discussed above, S364 and Q365 residues in the CUT and N476 in the HOX of OC2, form direct base-specific hydrogen bonds with the DNA. These residues are also conserved across OC, POU and SATB families (Fig. 2j) as well as evolutionarily, for example, S364 and particularly Q365 are conserved in the phage repressors[17,20] while N476 is conserved in yeast MATα2[41], and *Drosophila* Engrailed[42] and Antennapedia[43] transcription factors (Supplementary Fig. 4a, b) and across homeobox domains generally[22]. Accordingly, we generated two mutants, the first with S364A and Q365A mutations (OC2SQ; double mutant) and the second with N476A (OC2N) mutation. In addition, we also mutated the R479 and R480 (RR motif) to alanines (OC2RR; double mutant). The electron densities of the residues S364, Q365, N476, R479 and R480, as observed in our OC2-*PEG10* complex structure, are shown in Supplementary Fig. 4c.

We performed ITC-based DNA binding experiments with these three mutants and compared their thermodynamic parameters with that of wild-type OC2. Both OC2SQ and OC2N mutants bound weaker to *PEG10* DNA with a $K_D$ of 51 nM and 70 nM respectively (Table 3). Importantly, compared to wild-type OC2, both mutants showed a reduction in the respective enthalpy changes, by almost 30% ($\Delta H$ ~ −10 kcal/mol), while the entropy was more favorable (-T$\Delta S$ ~ 0 kcal/mol) in both cases (Fig. 4a, b; Table 3 and Supplementary Fig. 5b, c). These enthalpy and entropy values indicate weaker DNA binding and a disordered complex relative to wild-type OC2. Further, such a large shift in $\Delta H$ and -T$\Delta S$ cannot be solely accounted for by the localized loss of a few hydrogen bonds which suggest that these mutants, lacking proper DNA contacts, are unable to attain the right conformation upon binding to DNA. Notably, the mutated residues S364 and Q365 in OC2SQ are part of α3 helix that undergo structuring upon binding the DNA. The observed thermodynamic changes therefore suggest that the base interactions by S364/Q365 and N476 are essential for the conformational rearrangements in OC2. To test this further, we attempted to crystallize both OC2SQ and OC2N mutants with DNA. However, we failed to obtain any crystals of OC2N while with OC2SQ, we were only able to obtain poor quality crystals that were irreproducible, which might be indicative of the conformational variability and/or disorder in the respective complexes.

Next, we tested DNA binding by the OC2RR mutant and observed a similarly weaker binding ($K_D$ = 47 nM) (Table 3). Intriguingly, in contrast to the other two mutants, OC2RR neither showed a decrease in the enthalpy change ($\Delta H$ = −18.7 kcal/mol) nor a favorable change in entropy (-T$\Delta S$ = 8.7 kcal/mol) (Fig. 4c; Table 3 and Supplementary Fig. 5b, c) compared to the OC2-DNA complex. These values indicate a similar conformational state of this mutant in the DNA bound state like that of the wild-type OC2. To understand the role of the individual arginines, we introduced single alanine mutations at R479 and R480 (mutants OC2R479 and OC2R480) and tested their binding to *PEG10* DNA. OC2R479 mutant shows a slightly stronger affinity ($K_D$ = 6 nM) and modestly higher enthalpy change ($\Delta H$ = −15.9 kcal/mol) compared to the wild-type OC2 (Supplementary Fig. 5a−c). On the other hand, OC2R480, like the OC2RR double mutant, binds *PEG10* with a weaker affinity ($K_D$ = 47 nM) and shows lower enthalpy change ($\Delta H$ = −13.4 kcal/mol) (Supplementary Fig. 5a−c). These mutations were further analyzed using kinetics experiments (described in the next section). Additionally, to examine these residues, we crystallized OC2RR with the *PEG10* DNA and solved the structure at 2.9 Å resolution (Supplementary Table 1). This structure mostly resembles that of wild-type OC2-DNA complex (Supplementary Fig. 6). However, we could not model the few additional linker-flanking, and the last three C-terminal, residues located almost immediately after the RR mutation site, due to disorder (refer to methods for residue range). In addition, surprisingly, we could place only three water molecules in this structure. This might be due to the relatively lower resolution of the OC2RR-*PEG10* complex structure while may also suggest higher solvent disorder in the complex. Overall, these results indicate base interactions by this arginine pair stabilize the C-terminal of the protein and the overall complex.

Taken together, the above data suggest base interactions by S364/Q365, N476, and R479/R480 are needed for optimal DNA binding affinity. However, respective interactions by the evolutionarily conserved S364/Q365 and N476 are essential for accurate OC2 conformation required for favorable DNA binding energetics and are therefore mechanistically separable from that of OC-specific R479/R480.

### Base interactions by OC2, including the RR motif, are essential for optimal DNA binding and transcriptional activity

To further understand the interaction, we studied DNA binding kinetics of the wild-type and mutant OC2 proteins using biolayer interferometry (BLI). We observed that the association of wild-type

### Table 2 | Heat capacity analysis of OC2 and CUT binding to the *PEG10* DNA

| Interaction | Temp (°C) | ΔG (kcal/mol) | ΔH (kcal/mol) | -TΔS (kcal/mol) | ΔC° (cal/mol/K) |
|---|---|---|---|---|---|
| | 12 | −9.8 ± 0.5 | −12.5 ± 0.2 | 2.7 ± 0.5 | |
| OC2 - *PEG10* | 25 | −11.2 ± 0.2 | −15.5 ± 0.3 | 4.4 ± 0.2 | −440.7 ± 46.3 |
| | 30 | −10.9 ± 0.5 | −20.4 ± 1.0 | 9.5 ± 1.0 | |
| | 12 | −7.5 ± 0.2 | −2.1 ± 0.2 | −5.4 ± 0.3 | |
| CUT - *PEG10* | 25 | −7.8 ± 0.1 | −3.4 ± 0.2 | −4.4 ± 0.3 | −74.1 ± 6.8 |
| | 30 | −7.9 ± 0.1 | −3.4 ± 0.3 | −4.4 ± 0.4 | |

### Table 3 | ITC analysis of the binding of OC2 mutants to the *PEG10* DNA

| Interaction | $K_D$ (nM) | ΔG (kcal/mol) | ΔH (kcal/mol) | -TΔS (kcal/mol) | N (sites) |
|---|---|---|---|---|---|
| OC2SQ - *PEG10* | 51 ± 9 | −10.4 ± 0.7 | −10.7 ± 0.5 | 0.7 ± 0.5 | 0.9 ± 0.1 |
| OC2N - *PEG10* | 70 ± 11 | −9.8 ± 0.1 | −9.5 ± 0.4 | −0.2 ± 0.5 | 1.0 ± 0.1 |
| OC2RR - *PEG10* | 47 ± 8 | −10.0 ± 0.1 | −18.7 ± 0.5 | 8.7 ± 0.5 | 0.9 ± 0.3 |

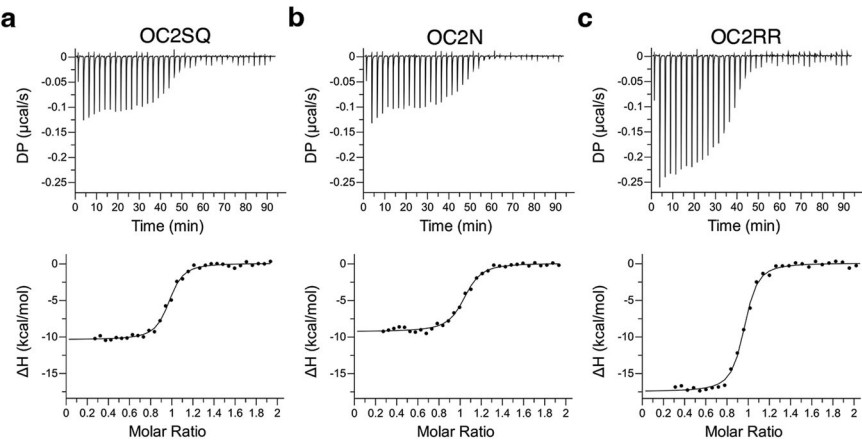

**Fig. 4 | DNA binding thermodynamics of OC2 base-specific mutants. a–c** ITC binding analysis of OC2SQ, OC2N and OC2RR, to *PEG10* DNA. The raw heats (differential power, DP) for each injection are shown on top and binding isotherms are shown in the bottom. The data shown is representative of three independent experiments (*n* = 3; technical replicates). Source data are provided as a Source Data file.

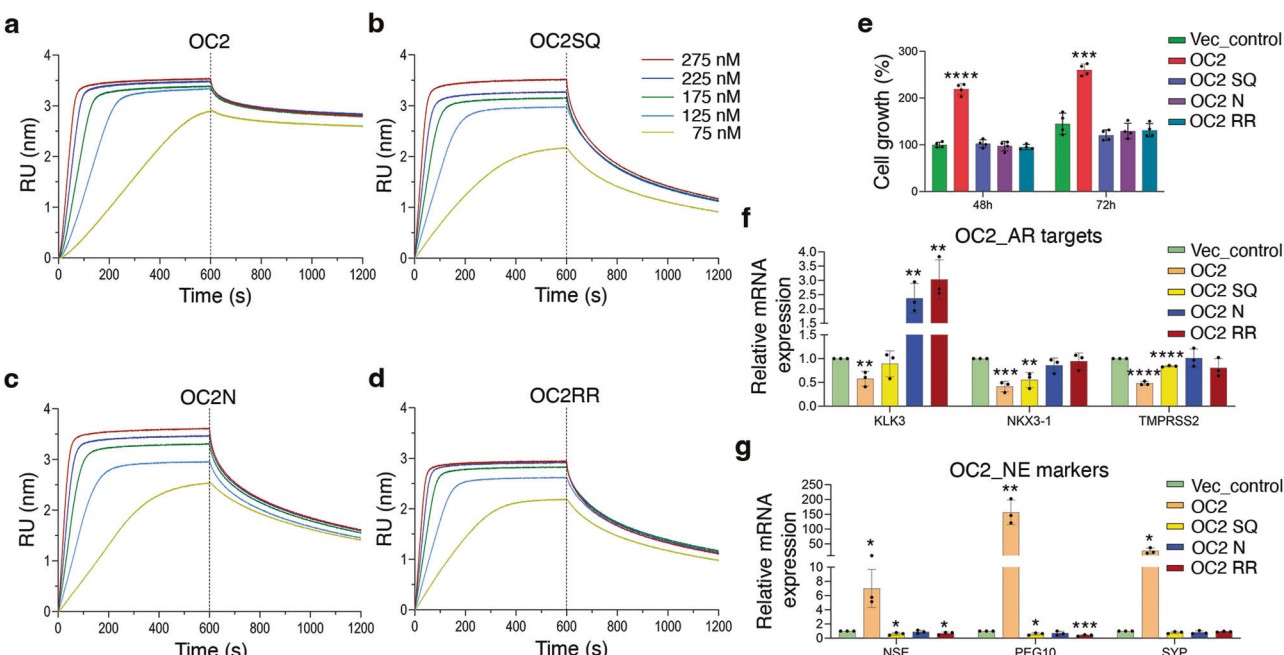

**Fig. 5 | Base-specific interactions by OC2 are needed for DNA binding cooperativity and are functionally relevant in a PC model. a–d** Kinetics of *PEG10* DNA binding by OC2, OC2SQ, OC2N and OC2RR. All proteins are titrated at concentrations 75, 125, 175, 225 and 275 nM, and representative curves are shown (*n* = 3; technical replicates). The association and dissociation phases are separated by a dotted line. **e** Plot showing the proliferation of LNCaP cells upon stable expression of ectopic wild-type and mutant OC2 compared to cells with endogenous OC2 (vector control) at 48 and 72 h. Data are presented as mean values ± SD. Two-sample *t*-test was used for statistical analysis (*n* = 4; biological replicates). At 48 h, ****P(OC2) < 0.0001, P(OC2SQ) = 0.64, P(OC2N) = 0.67, P(OC2RR) = 0.26, and at 72 h, ***P(OC2) = 0.00011, P(OC2SQ) = 0.09, P(OC2N) = 0.31, P(OC2RR) = 0.33. **f-g** Relative mRNA levels of AR target genes *KLK3*, *NKX3-1* and *TMPRSS2* and NEPC

marker genes *NSE*, *PEG10* and *SYP*, upon stable overexpression of ectopic wild-type and mutant OC2 compared to cells with endogenous OC2 (vector control). Data are presented as mean values ± SD. Two-sample *t*-test was used for statistical analysis (*n* = 3; biological replicates). In case of AR targets, for KLK3, **P(OC2) = 0.008, P(OC2SQ) = 0.53, **P(OC2N) = 0.009, **P(OC2RR) = 0.007; for NKX3-1, ***P(OC2) = 0.0007, **P(OC2SQ) = 0.008, P(OC2N) = 0.19, P(OC2RR) = 0.62; and for TMPRSS2, ****P(OC2) < 0.0001, ****P(OC2SQ) < 0.0001, P(OC2N) = 0.93, P(OC2RR) = 0.16. In case of NE markers, for NSE, *P(OC2) = 0.02, *P(OC2SQ) = 0.02, P(OC2N) = 0.45, *P(OC2RR) = 0.04; for PEG10, **P(OC2) = 0.003, *P(OC2SQ) = 0.01, P(OC2N) = 0.14, ***P(OC2RR) = 0.0008; and for SYP, *P(OC2) = 0.01, P(OC2SQ) = 0.09, P(OC2N) = 0.2, P(OC2RR) = 0.09. Source data are provided as a Source Data file.

OC2 to the DNA follows a sigmoidal curve characterized by a short lag phase or slower binding before optimal association commences (Fig. 5a and Supplementary Fig. 7d). This pattern is indicative of the cooperative nature of the association, like previously reported binding of bacterial protein ParA to DNA[44]. The binding is further characterized by a fast association ($k_a$) and slow dissociation ($k_d$) rates ($k_a$ = 320000 $M^{-1}s^{-1}$ and $k_d$ = 0.0005 $s^{-1}$; Table 4).

We next determined binding kinetics of the three mutants (OC2SQ, OC2N and OC2RR) to DNA. Interestingly, the association of the mutant proteins to the DNA lacked the sigmoidal pattern (Fig. 5b–d and Supplementary Fig. 7e–g) suggesting absence of the initial lag associated with the wild-type OC2. Furthermore, all three mutants dissociated faster, by one order of magnitude, relative to the wild-type OC2 (Table 4). We also tested binding kinetics of the OC2R479 and

**Table 4 | Kinetics analysis of wild-type and mutant OC2 proteins to the *PEG10* DNA**

| Interaction | $k_a$ (M$^{-1}$s$^{-1}$) | $k_d$ (s$^{-1}$) |
|---|---|---|
| OC2 - *PEG10* | 320000 ± 123000 | 0.0005 ± 0.0001 |
| OC2SQ - *PEG10* | 256000 ± 60100 | 0.0036 ± 0.0014 |
| OC2N - *PEG10* | 360000 ± 36700 | 0.0030 ± 0.0007 |
| OC2RR - *PEG10* | 162000 ± 101000 | 0.0028 ± 0.0010 |

OC2R480 single mutants to the DNA. In case of the OC2R479, the association curve retained the sigmoid shape but showed a moderately faster association and slower dissociation ($k_a = 560000$ M$^{-1}$s$^{-1}$; $k_d = 0.0004$ s$^{-1}$) compared to the wild-type OC2 (Supplementary Fig. 7a, c, h). On the other hand, the OC2R480, like the OC2RR mutant, lacked the sigmoid association and dissociated faster than the wild-type OC2 ($k_a = 694000$ M$^{-1}$s$^{-1}$; $k_d = 0.0048$ s$^{-1}$) (Supplementary Fig. 7b, c, i). These are consistent with the stronger and weaker binding affinities for OC2R479 and OC2R480, respectively, in ITC. As discussed in our structural analysis of the RR motif interactions, R480 interacts with a GC base-pair at position 6 of PEG10 that is totally conserved among OC promoters (Fig. 2a, b and Supplementary Fig. 2a, b). Accordingly, based on the sequence analysis, and our binding studies, this residue appears to be important for the DNA binding of OC family members. The faster binding kinetics of OC2R479, together with the ITC experiments indicating a slightly stronger binding of this mutant to the DNA compared to the wild-type OC2, might be indicative of this residue providing additional promoter specific structural rearrangements in OC2- relative to OC1-DNA complex, that is consistent with our structural analysis. The non-sigmoidal association pattern in the remaining mutants shows a loss in their DNA binding cooperativity. The kinetics further suggest that the base-specific interactions by the wild-type OC2 cause a slower association as well as dissociation, thereby stabilizing the complex. The kinetics data also show that the OC2RR mutant, despite exhibiting contrasting DNA binding thermodynamics than OC2SQ and OC2N mutants, is also defective in DNA binding.

Next, we wanted to validate whether these base-specific interactions are functionally relevant in terms of transcriptional activity and cancer cell proliferation using a cell-based PC model. Our earlier work and that by Guo et al.[28,29] showed that overexpression of OC2 leads to androgen receptor (AR) axis suppression and development of NEPC characteristics (lineage plasticity) in LNCaP cells, an AR-dependent prostate cancer model characterized by relatively lower endogenous OC2 expression. Upon constitutive overexpression of the OC2 SQ, N or RR mutant, instead of the wild-type OC2 protein, we found that the proliferation of the cells was reduced to that observed at endogenous OC2 levels (Fig. 5e). We analyzed the mRNA levels of three PC relevant AR target genes *KLK3*, *NKX3-1*, and *TMPRSS2*, in cells expressing either OC2SQ, OC2N or OC2RR mutants. Unlike the wild-type OC2, none of the three OC2 mutants could suppress these AR targets (Fig. 5f). Lastly, we tested expression of three NE differentiation markers *NSE*, *PEG10* and *SYP*, that are upregulated by OC2. Consistently, none of the mutants upregulated these genes (Fig. 5g).

In the case of the AR target *KLK3*, the OC2N and OC2RR mutants show a stronger effect than the endogenous OC2 (vector control), or in other words, a dominant negative effect. The reason for this observation is not precisely clear to us and might well be loci specific. However, we have recently shown that OC2 acts as a chromatin remodeler and can regulate promoter-enhancer contacts at the *KLK3* gene locus[45]. It is plausible that the CUT and HOX specific interactions with the DNA contribute differentially towards chromatin remodeling, which might explain the stronger effect of the HOX mutants we observed here in terms of the *KLK3* gene, although this needs further investigation.

In conclusion, our cell-based assays validate the interactions we have identified and characterized biochemically to be necessary for cell-proliferation and OC2 transcriptional activity in the prostate cancer model tested.

## Discussion

The homeodomain (HOX) is a ubiquitous gene regulatory element that can combine with CUT domain(s) to constitute the CUT class of transcription factors[46,47]. The CUT-HOX combination constitutes the DNA binding domain of several transcription factor families, including OC, SATB and CUX and the closely related POU, that regulate various developmental and housekeeping pathways. Despite widespread occurrence and fundamental biological roles of the CUT-HOX module, its DNA binding mechanism is not well understood. While previous structural analyses of POU members and OC1 provided key initial insights into the positioning of the CUT and HOX domains on DNA, these studies reveal little information about their coordinated binding mechanism. Here, we report an integrative analysis of the DNA binding of OC2, a member of the OC family, that is also a driver and therapeutic target of treatment-resistant prostate cancer. We show that the CUT domain, unlike HOX, can bind DNA on its own albeit weakly. However, the HOX domain is critical in driving an energetically favorable OC2-DNA complex by allosterically inducing rearrangements in the CUT domain. This implies a two-step mechanism of cooperative DNA binding by OC2 wherein initial contacts to DNA are made by CUT followed by binding of HOX that thermodynamically stabilizes OC2 onto DNA. In parallel structural studies, we identified a unique DNA base interacting arginine pair in the HOX domain of OC2, which we call the 'RR motif'. This amino acid pair is unique to the OC family compared to POU and SATB. In addition, the first arginine interacts distinctly to DNA in respective OC2-*PEG10* and OC1-*TTR* complexes, suggesting a mechanism to tolerate specific alterations in OC promoter sequences with implications on the redundant transcriptional activation by OC paralogs[10,27]. These findings together demonstrate the HOX domain to be a key regulatory element for OC2-DNA binding.

Probing the mechanism further, we discovered that DNA base contacts by S364/Q365 in CUT and N476 in HOX, residues conserved evolutionarily and across OC, POU and SATB families, to be essential determinants for an energetically favorable and therefore, conformationally correct OC2-DNA complex. Nonetheless, base interactions by OC specific R479/R480 (RR motif), apart from S364/Q365 and N476, are needed for optimal DNA binding affinity, kinetics, and cooperativity. Notably, in a prostate cancer model, we show these interactions to be essential in terms of OC2 transcriptional activity and cancer cell proliferation. Collectively, these findings demonstrate that the respective DNA interactions by the evolutionarily conserved amino acids S364/Q365 and N476 ensure the basic functional framework while family-specific elements in the HOX domain, like the RR motif in OC, provide additional mechanistic properties to the OC family.

In conclusion, we propose that the OC2 HOX domain, with its crucial thermodynamic contribution and unique RR motif, regulates stable OC2-DNA interaction. Further, considering the prevalence of the HOX domain in transcription factors, its thermodynamic contribution towards DNA binding might be of broader significance. In addition, the HOX-induced conformational change in the CUT domain, needed for driving a thermodynamically favorable interaction with the DNA, could be therapeutically relevant. For instance, CD4-induced rearrangement in the HIV gp120 has been harnessed for development of potent antivirals[48,49]. Finally, the unique interactions of the RR motif in the OC2 HOX domain relative to corresponding amino acids in OC1 and POU members, which share an otherwise conserved HOX domain in terms of both sequence and structure, reveal a specific vulnerability for targeting of OC2. These findings might be relevant in the context of strategies being constantly sought to target transcription factors, often considered 'undruggable'[50]. Overall, our integrative approach

reveals molecular details of DNA binding by OC2 with broad mechanistic implications for CUT and related POU family transcription factors, and that present potential therapeutic opportunities for intervention.

## Methods

### Protein expression and purification
The human OC2 DNA binding region spanning residues 330–485 (OC2) was cloned into pET-His6-TEV-LIC expression plasmid (Addgene Plasmid #29653). The protein was expressed in *Escherichia coli* (*E. coli*) BL21(DE3) cells. The cells were grown at 37 °C to an optical density (OD) of 0.8 in Terrific Broth (TB) media and induced with 0.5 mM isopropyl β-D-1-thiogalactopyranoside (IPTG) at 18 °C overnight. Cells were lysed by sonication in buffer containing 50 mM Tris pH 7.5, 500 mM NaCl, 20 mM Imidazole, 10% Glycerol and 5 mM β-mercaptoethanol (β-ME) (Buffer A). Cell debris were removed by centrifugation at 43,600 x *g* and cleared lysate was passed through nickel-nitrilotriacetic acid (Ni-NTA) resin (Qiagen). The protein was eluted in Buffer A supplemented with 500 mM Imidazole. The His-tag was removed by incubating Ni-NTA eluate with Tobacco Etch Virus (TEV) protease at 4 °C overnight. The sample was diluted to reduce NaCl and imidazole concentrations to 50 mM each and passed through Ni-NTA resin again to remove the cleaved His-tag and TEV (also His-tagged). The protein was then loaded on a 5 mL HiTrap SP (Cytiva) cation exchange column, equilibrated in buffer containing 25 mM HEPES pH 7.4, 50 mM NaCl, 10% Glycerol and 1 mM DTT and eluted with a linear gradient of 50 mM to 1 M NaCl. The fractions containing OC2 were concentrated and loaded onto a Superdex S75 gel filtration column (Cytiva) equilibrated with buffer containing 25 mM HEPES pH 7.5, 250 mM NaCl and 1 mM DTT. The purified protein was aliquoted, flash-frozen in liquid nitrogen and stored at −80 °C. All the mutants were prepared using the same protocol. The final purified wild-type and mutant OC2 showed similar SDS-PAGE and gel-filtration elution profiles (Supplementary Fig. 8). The respective elution profiles were plotted using GraphPad Prism. The OC2 residues 317-417, containing the CUT domain, and residues 420–490, containing the HOX domain were cloned into pET-His6-MBP-TEV-LIC expression plasmid (Addgene Plasmid #29656). Both proteins were purified using the same protocol described above for the intact OC2 protein. Sequences of all oligonucleotides used have been provided in Supplementary Table 3.

### Site-directed mutagenesis
Mutations in OC2 DNA binding region (residues 330–485) for purified protein-based studies were introduced by site-directed mutagenesis using Pfu Turbo (Agilent) DNA polymerase. The PCR product was treated with DpnI (NEB) enzyme at 37 °C for 1 h and transformed into Top10 *E. coli* cells. Mutagenesis in full length *OC2* for the cell-based assays were performed using Quick Change II XL site-directed mutagenesis kit (Agilent) according to manufacturer's protocol. Mutations were confirmed by DNA sequencing.

### Crystallization, data collection and structure determination
OC2 DNA binding site was originally mapped to 14 base pairs within *PEG10* promoter sequence[28], so, we initially attempted to co-crystallize OC2 with the corresponding 14 mer DNA duplex (Supplementary Fig. 1e). However, this 14 mer DNA yielded crystals that were difficult to reproduce. Changing the DNA to a 12 mer duplex, lacking one base pair from each terminus in comparison to the 14 mer sequence, resulted in crystals that formed more readily. DNA oligos (IDT) were annealed for crystallization and duplex DNA formed was mixed with protein in 1:1.4 ratio (protein to duplex DNA). Crystallization was set-up at 18 °C by hanging drop vapor diffusion method. OC2-*PEG10* complex crystals were obtained in the condition 0.04 M KH₂PO₄, 16 % PEG 8000 and 20 % Glycerol while OC2RR-*PEG10* complex crystals appeared in the

condition 10 % PEG 1000 and 7.5 % PEG 8000. Data were collected in an in-house Rigaku Micromax 007 HF rotating anode X-ray generator and R-axis IV + + image-plate detector. Data processing was performed with HKL2000[51]. Structure determination of OC2-*PEG10* was carried out by molecular replacement method using MolRep[52], with the OC1-*TTR* complex structure (PDB 2D5V) as a search model. For the OC2RR-*PEG10* structure solution, OC2-*PEG10* structure was used as a search model. Model building was done with COOT[53] while refinement was carried out using REFMAC[54,55] and Phenix Refine[56]. In both structures, the amino acids 409-428, representing the linker, could not be modeled due to lack of electron density. In addition, the OC2RR-*PEG10* structure also lacked proper electron density for residues 407-408, 429-432 and 483-485. The data collection and refinement statistics are provided in Supplementary Table 1. Structure figures were prepared with PyMOL (The PyMOL Molecular Graphics System, Version 2.4 Schrödinger, LLC). Structural alignments and respective rmsd calculations were also performed using PyMOL. All the above crystallographic softwares were used from the SBgrid platform[57]. Protein-DNA interaction map was prepared with LigPlot+[58]. Distances between DNA phosphate backbones were calculated using 3DNA[59].

### ITC binding studies
ITC experiments were performed using MicroCal PEAQ-ITC (Malvern Panalytical). Both protein and DNA were dialyzed in 1X phosphate buffered saline (PBS) pH 7.4, 0.005% Tween-20 and 1 mM β-ME. For experiments involving intact (wild-type and mutant) OC2, duplex DNA at 100 µM (in syringe) was titrated as 36 injections of 1 µL each against 10 µM protein (in the cell). For experiments involving isolated CUT and HOX domains, approximately three to five-fold higher concentrations of protein and DNA were used due to the lower heats generated by the domains when separated. Accordingly, for these experiments, DNA at higher concentrations (300 µM and 500 µM for experiments involving CUT and HOX, respectively) (in the syringe) was titrated as 18 injections of 2 µL each against respective proteins (30 µM CUT and 50 µM HOX) (in the cell). All experiments were carried out in triplicate ($n = 3$; technical replicates) and the data were processed with MicroCal PEAQ-ITC Analysis software. Enthalpy change (ΔH) at temperatures 12, 25 and 30 °C were calculated and plotted. The slope of this graph yielded the heat capacity (ΔC°; enthalpy change per mole per unit temperature (cal/mol/K)). Figure panels depicting raw heats (DP) and binding isotherms were generated from MicroCal PEAQ-ITC Analysis software. The bar graphs showing signature plots were prepared in GraphPad Prism. All data are represented as mean values ± SD of three technical replicates. Significance analyses were performed using one-way ANOVA.

### Biolayer interferometry (BLI) kinetics studies
Protein-DNA kinetic studies were carried out in Octet RED96 (Sartorius ForteBio). One of the *PEG10* oligos was biotinylated on the 5′-end (IDT) and annealed to the non-biotinylated complimentary oligo. This biotinylated duplex DNA was immobilized on a SADH biosensor (Sartorius) and OC2 (wild-type or mutants) was titrated at 0, 75, 125, 175, 225 and 275 nM concentrations. The assays were performed in 1X phosphate buffered saline (PBS), 0.5 mM tris [2-carboxyethyl] phosphine (TCEP) and 0.005% Tween-20. The 75 nM curve showed poor fitting, so, we used the 125 to 275 nM curves for calculating the kinetic parameters. All data are represented as mean +/- SD of three technical replicates. Data was fitted with 1:1 model using TraceDrawer software (Ridgeview Instruments). Significance analyses were performed using one-way ANOVA. Figure panels depicting binding kinetics were prepared in GraphPad Prism.

### HDX-MS data collection and analysis
HDX-MS was performed on a Waters HDX-1 system, which consists of a Leap autosampler (Leap Technologies, Carrboro, NC), coupled to a

Synapt G2-Si Qtof mass spectrometer (Waters Corporation, Milford, MA). $D_2O$ buffer was prepared by lyophilizing sample buffer (10 mM $Na_2HPO_4$, 1.8 mM $KH_2PO_4$, 137 mM NaCl, 2.7 mM KCl, 0.5 mM TCEP pH 7.4), then redissolving it in an equivalent volume of 99.9% $D_2O$ (Cambridge Isotope Laboratories, Andover, MA). Proteins (5 μM final) were combined with sample buffer (Control) or PEG10 ( + DNA) at a final concentration of either 7.5 μM (OC2) or 200 μM (CUT) in a final volume of 150 μL. After 15 min at room temperature (RT), samples were held at 1 °C until dispensing: 4 μL was transferred to a 25 °C tube, equilibrated for 5 min before mixing with either $H_2O$ (control) or $D_2O$ buffer (56 μL) for the indicated times (0 min, 0.25 min, 0.5 min, 1 min, 2 min). 50 μL of the $H_2O$- or $D_2O$-incubated sample was then transferred to a 1 °C tube containing 50 μL 3 M guanidine hydrochloride (final pH 2.66) and incubated for 1 min to quench deuterium exchange and denature the protein prior to injection of 90 μL into an in-line 15 °C pepsin column (Immobilized Pepsin, Pierce). Peptides were captured on a BEH C18 Vanguard precolumn then separated by analytical chromatography (Acquity UPLC BEH C18, 1.7 μm 1.0 × 50 mm, Waters Corporation) over 7.5 min using a 7–85% acetonitrile gradient before electrospray into the Synapt G2-Si. Data were collected in the Mobility, ESI+ mode using an acquisition range of 200–2000 m/z and scan time of 0.4 secs with leu-enkephalin (m/z = 556.277) as lock mass (mass accuracy, 1 ppm)[60].

To identify peptides, the Synapt was run in mobility-enhanced data-independent acquisition (MSE), mobility ESI+ mode. Peptide masses were determined from triplicates and analyzed using ProteinLynx global server (PLGS) v3.0 (Waters Corporation) using cutoffs of 250 ion counts for low energy peptides, 50 ion counts for fragment ions, and 1,500 Da minimum mass. PLGS-identified peptides were processed with DynamX v3.0.0 (Waters Corporation) by comparing mass envelope centroids[61]. Data are represented as mean +/- SD of three technical replicates. Deuterium loss was corrected using a global back exchange factor determined from the average exchange measured in disordered termini of varied proteins[62]. Significance among differences was assessed using ANOVA and t-test (P-value < 0.05) using DECA (v116)[63] (github.com/komiveslab/DECA). Structural representations, including uptake maps were prepared using PyMol. The α3 helix uptake plot was obtained from DECA.

## Protein sequence alignments
Protein sequence alignments were performed using Clustal Omega[64] and edited in Jalview[65]. Respective alignment images were exported from Jalview.

## Stable cell line generation
LNCaP (#CRL-1740) was obtained from the American Type Culture Collection (ATCC) and authenticated using the Promega PowerPlex 16 system DNA typing (Laragen). Mycoplasma contamination was routinely monitored using the MycoAlert PLUS Mycoplasma Detection Kit (Lonza). The OC2 overexpression construct was generated by cloning the full-length OC2 cDNA (NM_004852) into the pLenti-C-Myc-DDK-IRES-Puro (Origene PS100069) lentivirus system. Then packing (psPAX2, Addgene #12260), and envelope (pMD2.G, Addgene #12259) plasmids were co-transfected into HEK293T cells to produce lentivirus. Cells were infected with lentivirus supplemented with 10 μg/mL polybrene, then selected by 2 ug/mL puromycin to generate the stable overexpression cells. All cell lines were grown in RPMI-1640 media (Gibco) supplemented with 10% FBS and penicillin/streptomycin.

Relative mRNA expression levels of endogenous *OC2* (vector control), wild-type *OC2*, *OC2SQ*, *OC2N* and *OC2RR* in respective stably expressing LNCaP cells are shown in Supplementary Fig. 7 m.

## Cell proliferation analysis
All procedures were performed according to the XTT cell viability kit protocol (CST). Seeding was done with 2000 cells/well and grown up to 72 h, then absorbance at 450 nM was measured for further analysis.

Assays were performed in triplicates ($n = 3$; biological replicates) and significance analysis was performed using two sample t-test.

## RT-qPCR for gene expression analysis
Total RNA from cells was extracted using Qiagen RNeasy Kit (Qiagen) following the manufacturer's instructions. 1 μg of total RNA was reverse transcribed to cDNA with iScript cDNA Synthesis Kit (Bio-Rad) following manufacturer's instructions. 2X PowerUp SYBR Green Master Mix (ThermoFisher) was used for cDNA amplification. Assays were performed in triplicates ($n = 3$; biological replicates) and normalized to β-actin. Significance analysis was performed using two sample t-test.

## Graphs
Graphs were prepared using GraphPad Prism (www.graphpad.com) where indicated.

## Statistics and reproducibility
Statistical analyses details have been provided in respective figure legends and methods.

## Reporting summary
Further information on research design is available in the Nature Portfolio Reporting Summary linked to this article.

## Data availability
The crystallographic data with the PDB accession codes [8T0F] (OC2-*PEG10*) and 8T11 (OC2RR-*PEG10*) are available at wwpdb.org. HDXMS data is available at massive.ucsd.edu. (Dataset MSV000094672 [10.25345/C5KH0F95S]). Source data are provided with this paper. The following prior published structures used for analyses in this work are available at wwpdb.org: 2D5V (OC1-*TTR*); [1S7E] (apo OC1); 1E3O (OCT1); 1AU7 (PIT1); 2OR1 (434 phage repressor); 1LMB (Lambda phage repressor); 1APL (yeast MATα2); 1HDD (Engrailed); 9ANT (Antennapedia). Source data are provided with this paper.

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

## Acknowledgements

The authors thank staff at Functional Genomics Core at University of Arizona, Tucson and X-ray and EM structure determination core at UCLA for granting access to Octet instrument. This work was supported by National Institutes of Health (1R01CA220327, 2P50CA092131) grants to M.R.F. and US Department of Defense (PC210486) grant to M.R.F. and R.M.. B.G. is supported by a National Cancer Institute grant (T32CA240172). The HDX core of the UCSD Biomolecular Proteomics Mass Spectrometry Facility is supported by NIH shared instrumentation grant number S10 OD016234.

## Author contributions

R.M., M.R.F. and A.C. designed research; R.M. and M.R.F. supervised the work. A.C. performed protein purifications, crystallization, structure analysis, mutant design, and kinetics studies. M.K. and A.C. carried out structure determination and refinement. B.G. performed ITC experiments. M.K., M.R.H., and B.G. helped with protein purifications. C.Q. generated stable cell-lines, performed cell-proliferation and gene expression analyses. S.S. & E.A.K. performed HDX-MS experiments and data analysis. A.C. wrote and edited the original draft with critical reading by M.K., B.G., R.M. and M.R.F. and inputs from all authors. M.R.F. and R.M. acquired the funding.

## Competing interests

The authors declare no competing interests.
