## [Peer Review File · Nature Communications]

REVIEWER COMMENTS

Reviewer #1 (Remarks to the Author):

Chatterjee et al. report the crystal structure of the DNA-binding domain of a transcription factor ONECUT2 with a target dsDNA, which identifies key residues for its sequence-specific DNA-binding and shows that the tandem DNA-binding (CUT and HOX) domains linked by a flexible linker bind to adjacent sites in the DNA major groove. The structural data are accompanied by corroborating biochemical and cellular data. Overall, the structural data are sound but of limited novelty given that the structure is essentially identical to that of the OC1-TTR complex (PDB 2D5V), and the biochemical data need more thorough analyses to support the major conclusions.

The authors show that, even though the HOX domain by itself shows no affinity for the PEG10 promoter DNA, the CUT and HOX domains bind to this DNA cooperatively. The authors suggest that the HOX domain allosterically modulates CUT to stabilize the complex. However, this model seems inconsistent with the crystal structure, in which the two domains do not interact with one another and the linker connecting them is disordered. How reliable are the ITC data in Extended Fig. 3? I find it a little bit hard to explain that the homeodomain by itself shows no binding at all to either DNA or CUT and yet greatly stimulates CUT binding to the DNA.

The BLI data are misinterpreted. The data should be fit to a 1:1 model, not 1:2. Introducing two association/dissociation rates would allow a better fit (simply by introducing more parameters, curve fitting would always work better) but there should not be two OC2 molecules bound to a PEG10 DNA, based on the crystal structure.

The 2nd Arg in the "RR" motif is a conserved residue in homeodomains, whose base-specific interaction has been documented in many studies in the literature. Thus, the observation should be discussed in the context of prior knowledge. The first Arg is more unique to this family. It would have been interesting to individually mutate these two Arg to Ala, rather than mutating both at the same time (as in OC2RR), to investigate the contribution of each residue.

There should be figures showing the electron density for key amino acids, especially given the modest crystallographic resolution (2.6 and 2.9 Å) of the present work.

Line 82: A bulge in the major groove sounds unclear in meaning.

Line 267: Fewer water molecules were observed most likely simply due to a lower resolution.

Reviewer #2 (Remarks to the Author):

This contribution describes the structure of the ONECUT2-DNA complex. Based on the structural findings, several mutants were created and characterized biophysically, in terms of their binding thermodynamics and kinetics. Overall the manuscript is well written and illustrated. Hence, the remaining review mainly investigates major claims expressed, based on the data provided.

Allostery by structural changes: comparison of ONECUT1 in the absence of DNA (referring to reference 29, where it was named HNF-6) and ONECUT2 in the presence of DNA (this contribution) reveals a relative minor reorientation of helices in the respective CUT domains. The significance of this change is relatively weak, given that the structures have been determined in different ways (NMR, X-ray). Quantification of these differences would be useful, to allow better evaluation to what extent these changes exceed expected experimental positional errors. As the argument on allostery is built on possible conformational changes, any further interpretation largely depends to what extent further evidence on the significance on these alterations can be provided.

In light of highlighting the importance of the RR motif in this contribution, the characterization of the OC2RR mutant is critical. Unfortunately, there is no explicit information on what kind of mutation is behind the OC2RR variant. Please clarify this (I assume it's AA, as this was also used for other mutants). The structural data of the OC2RR reveal a structure literally identical to the wt structure (Extended Figure 4). The ITC data of the mutant (Figure 4) show an approximately 5-fold decrease of binding affinity, similar to other mutants analyzed (OC2N, OC2SQ). Similarly, the kinetic data show an increase of the DNA dissociation rate (Figure 5). Unfortunately, both the thermodynamic and kinetic data are presented without repeated experiments and significance analysis. Therefore, judging the significance at present is a matter of guessing, and proper addition of such analysis is requested.

Judging to what extent the data shown in Figure 5f-h is somewhat out of my own core expertise. Nevertheless, I am wondering whether comparing endogenous (vector control) versus different ONECUT2 version over expressed is still state-of-the-art (Figure 5f). Why was it not possible to carry such experiments by complementation using a KO version as background? PCR of KLK3 shows a much stronger effect for the OC2N and OC2RR mutants than for the OC2SQ mutant (Figure 5g). This effect is not discussed in the text. Overall, as the last paragraph of the results section

addresses potential functional effects resulting from the mutations carried out, I would have wished to see a more in-depth functional characterization, followed by mechanistic explanations.

In conclusion, the paper is interesting and addresses an important question on allosteric mechanisms by binding of multiple protein domains to DNA, ultimately generating strong constitutive DNA motif binding. However, the manuscript in its present form falls short to explain any underlying allosteric mechanism, beyond the descriptive part of the manuscript. It would be very useful to achieve a better understanding on what kind of conformational changes, passing significance tests, are associated.

Additional specific comments:

In the abstract it is stated that “evolutionarily conserved base interactions additionally determine the thermodynamic stability of the ONECUT2-DNA complex”. I guess what the author are referring to is “strength of DNA binding” or similar. “Thermodynamic stability” implies increased temperature-dependent stability, which was not measured in this contribution.

Introduction: “OC2 is involved in other cancer types, including breast cancer, where it similarly acts to drive lineage plasticity (manuscript in review).” Please provide this manuscript as additional source for review or publish it in an accessible way (Biorxiv).

Figure 2

Panel j: I guess the alignment has directly been taken from Jalview. I would have preferred an alignment, which focuses more on key features discussed in this paper. Please explain the different color codes, or refer to a reference where these color codes are defined. It is of note that the RR motif does not belong to the most conserved residues within the alignment.

Figures 3 and 4: The original ITC data could be moved into the Supplement. Proper statistical and significance analysis is missing. The histogram presentation and inserted table show the same content. Move one of the two into Supplement. It would be straight forward to merge the two figures.

Figure 5: Proper statistical and significance analysis for data shown in panels a-d is missing.

Extended Figure 1: Quantification of structural differences would be useful.

Extended Figure 3: This figure can be dropped as it only shows negative data.

Supplementary Table 1: Please provide values with realistic precision (unit cell, Wilson B factor, R merge, R meas etc.)

We thank the reviewers for their valuable time and for providing thoughtful critiques of our
manuscript titled “The homeodomain regulates stable DNA binding of prostate cancer target
ONECUT2”. We have addressed the points raised by reviewers as comprehensively as possible.
Briefly, we performed the binding experiments in triplicates and have presented the data with relevant
statistics. To resolve the role of each arginine of the RR motif, we have mutated them individually (in
addition to the previous double arginine mutation) and tested the binding of the respective single
mutants to DNA. To characterize the structural changes upon DNA binding, first, we determined the
heat capacity for DNA binding by OC2 and CUT by ITC and second, we performed hydrogen-
deuterium exchange mass spectrometry (HDX-MS). We have accordingly modified our main text
(highlighted in GREEN text). We believe these additional experiments further support the conclusions
drawn and address the concerns of the reviewers. Please find a point-by-point response to the
reviewers’ comments below:

**Reviewer #1 (Remarks to the Author):**

*Chatterjee et al. report the crystal structure of the DNA-binding domain of a transcription factor*
*ONECUT2 with a target dsDNA, which identifies key residues for its sequence-specific DNA-binding*
*and shows that the tandem DNA-binding (CUT and HOX) domains linked by a flexible linker bind to*
*adjacent sites in the DNA major groove. The structural data are accompanied by corroborating*
*biochemical and cellular data. Overall, the structural data are sound but of limited novelty given that*
*the structure is essentially identical to that of the OC1-TTR complex (PDB 2D5V), and the biochemical*
*data need more thorough analyses to support the major conclusions.*

- 1. The authors show that, even though the HOX domain by itself shows no affinity for the PEG10
promoter DNA, the CUT and HOX domains bind to this DNA cooperatively. The authors suggest
that the HOX domain allosterically modulates CUT to stabilize the complex. However, this model
seems inconsistent with the crystal structure, in which the two domains do not interact with one
another and the linker connecting them is disordered.

Response:

We appreciate this question and would like to clarify this further. Indeed, the HOX and CUT do not
interact, as pointed out here. We have used the term ‘allosteric’ because HOX-DNA interaction
induces conformational changes in the $\alpha 3$ helix (**Extended Data Fig. 1d**), a site that is distal from this
interaction, in the CUT domain, and without involving physical CUT-HOX binding. The above
observations therefore are consistent with an allosteric mechanism.

- 2. *How reliable are the ITC data in Extended Fig. 3? I find it a little bit hard to explain that the*
*homeodomain by itself shows no binding at all to either DNA or CUT and yet greatly stimulates*
*CUT binding to the DNA.*

Response:

We have repeated all the binding experiments, including ITC, in exact triplicates. Additional replicates
for the HOX/DNA and CUT+HOX/DNA ITC experiments were also carried out to ensure these results
were reliable. Consistent with our initial results, even after repeated attempts, we have not noticed any
binding of the isolated HOX domain to the DNA in ITC. In addition, we have also verified the lack of

HOX-DNA binding using kinetics experiments but did not observe any binding here as well
(**Extended Data Fig. 7j**). Notably, other groups also reported a lack of binding between the OC HOX
and DNA using gel-shift assays (Lannoy et al.¹; <https://doi.org/10.1074/jbc.273.22.13552>). These
results support the conclusion that the HOX domain on its own does not possess appreciable DNA
binding in the absence of CUT. Mechanistically, we propose that upon initial contacts by CUT
concomitantly bringing HOX in proximity to the DNA, the two domains crosstalk through the DNA to
stabilize the complex.

3. *The BLI data are misinterpreted. The data should be fit to a 1:1 model, not 1:2. Introducing two*
*association/dissociation rates would allow a better fit (simply by introducing more parameters,*
*curve fitting would always work better) but there should not be two OC2 molecules bound to a*
*PEG10 DNA, based on the crystal structure.*

Response:

We have now fitted the BLI data using a 1:1 binding model. We have performed each experiment in
exact triplicate and modified the table (**Fig. 5**). The results are overall consistent with the initial version
of our manuscript showing a loss of sigmoid association and significantly faster dissociation of the
mutants.

4. *The 2nd Arg in the “RR” motif is a conserved residue in homeodomains, whose base-specific*
*interaction has been documented in many studies in the literature. Thus, the observation should be*
*discussed in the context of prior knowledge. The first Arg is more unique to this family.*

Response:

This is an important point. Among homeodomains of other families (non-OC), in general, the positions
corresponding to the RR motif we have studied in this manuscript, show considerable variability. Some
of these homeodomains do have an arginine at one of these two positions but not at both
simultaneously. For example, MAT α has an arginine at the first position (referred to as R54 in the
review by Gehring et al.; [doi: 10.1016/0092-8674\(94\)90292-5](https://doi.org/10.1016/0092-8674(94)90292-5)) but not in the second, which is a
lysine². In the related Engrailed (Eng) homeodomain, these arginine residues are substituted by an
alanine followed by a lysine (**Reviewer Fig. 1**). In Antp, another related homeodomain containing
protein, these residues are a methionine followed by a lysine. The arginine in MAT α and methionine in
Antp do interact with the DNA in base-specific manner but the second residue does not display base-
specific interaction in these proteins. Similarly, among POU members, only PIT1 has an arginine at the
second position but it does not interact in base-specific manner. This discussion was already in the
manuscript text (**Page 5; Lines 153-165**) and in addition, we have further modified our main text to
include information we discussed above about corresponding residues in MAT α , Eng and Antp (**Page**
**6; Lines 172-179**). Together, however, the above observations do suggest that the conserved RR pair
in OC is unique to this family.

```
OC2HOX/429-484  S R L V F T D L Q R R T L F A I F K E N - - - K R P S K E M Q I T I S Q Q L G L E L T T V S N F F M N A R R R S L E K  
MATa2HOX/41-99  R G H R F T K E N V R I L E S W F A K N I E N P Y L D T K G L E N L M K N T S L S R I Q I K N W V S N R R K I E K T I  
EngHOX/457-512  P R T A F S S E Q L A R L K R E F N E N - - - R Y L T E R R R Q Q L S S E L G L N E A Q I K I W F Q N K R A K I K K S  
AntpHOX/300-355 G R Q T Y T R Y Q T L E L E K E F H F N - - - R Y L T R R R R I E I A H A L C L T E R Q I K I W F Q N R R M K W K K E
```

**Reviewer Fig. 1:** Multiple sequence alignment of HOX domains of OC2, MAT α , Eng and Antp. The RR motif
in OC2HOX and corresponding residues in the other proteins are shown in a red outline. Residue ranges for the
respective HOX sequences are noted.

5. *It would have been interesting to individually mutate these two Arg to Ala, rather than mutating both at the same time (as in OC2RR), to investigate the contribution of each residue.*

Response:

According to this suggestion, we have made two single mutants R479A and R480A for DNA binding analysis. In ITC, the R480A mutant, shows a weaker binding affinity (KD ~47 nM), like the OC2RR double mutant (**Extended Data Fig. 5**). On the other hand, R479A mutation showed a moderate yet reproducibly stronger affinity relative to the wild-type OC2, to the DNA. To further investigate this observation, we studied the kinetics of both the single mutants. R480A mutant, as expected, showed a loss of the sigmoid association and faster dissociation, much like the OC2RR double mutant. However, interestingly the R479A mutant shows a moderately faster association and slow dissociation kinetics compared to the wild-type OC2 (**Extended Data Fig. 7a,b,c,h&i**).

As was discussed in the manuscript, the R480 residue interacts with a totally conserved GC base pair at position 6 of PEG10 (**Page 5; Lines 146-151 and Extended Data Fig. 2a**) and seems to be important for the DNA binding affinity of OC family, and the above results further support this notion. The contrasting observations with R479A mutant, although not as drastic, suggest that this residue might be important for specific structural adjustments required for optimal binding affinity/kinetics of the OC paralogs to the cognate promoter DNA sequences. This is also supported by our structural observations that this residue can re-orient based on the promoter sequence. These results have been discussed in **Page 8; Lines 294-300** and **Page 9; Lines 333-347**.

6. *There should be figures showing the electron density for key amino acids, especially given the modest crystallographic resolution (2.6 and 2.9 Å) of the present work.*

Response:

We have added a figure showing the electron densities for S364, Q 365, N476, R479 and R480 (**Extended Data Fig. 4 and Page 8; Lines 268-269**).

7. *Line 82: A bulge in the major groove sounds unclear in meaning.*

Response:

We have quantified the differences and have included a table (**Extended Data Fig. 1b**) listing the differences between the OC2-bound PEG10 and a canonical B-DNA. We have also edited the manuscript text (**Page 4; Lines 83-84**).

8. *Line 267: Fewer water molecules were observed most likely simply due to a lower resolution.*

Response:

We agree that the lower resolution might be a possible reason and have modified the text accordingly (**Page 9; Line 306-308**).

**Reviewer #2 (Remarks to the Author):**

*This contribution describes the structure of the ONECUT2-DNA complex. Based on the structural*
*findings, several mutants were created and characterized biophysically, in terms of their binding*
*thermodynamics and kinetics. Overall, the manuscript is well written and illustrated. Hence, the*
*remaining review mainly investigates major claims expressed, based on the data provided.*

Response:

We thank the reviewer for the positive comments.

1. *Allostery by structural changes: comparison of ONECUT1 in the absence of DNA (referring to*
*reference 29, where it was named HNF-6) and ONECUT2 in the presence of DNA (this*
*contribution) reveals a relative minor reorientation of helices in the respective CUT domains. The*
*significance of this change is relatively weak, given that the structures have been determined in*
*different ways (NMR, X-ray). Quantification of these differences would be useful, to allow better*
*evaluation to what extent these changes exceed expected experimental positional errors. As the*
*argument on allostery is built on possible conformational changes, any further interpretation*
*largely depends to what extent further evidence on the significance on these alterations can be*
*provided.*

Response:

We appreciate this comment and accordingly carried out new experiments - 1) we calculated the heat
capacity (ΔC°) of the binding to further verify the conformational change (**Page 7, Lines 222-234; Fig.**
**3e and Extended Data Fig. 3b**). Negative ΔC° indicates ligand-dependent protein-folding^{3,4}. Based on
our experiments, we calculated a ΔC° of -440 cal/mol/K for the OC2-DNA interaction. In similar
experiments for binding of the isolated CUT domain to the DNA, we observed a much lower ΔC° of -
74 cal/mol/K, indicating a lack of conformational change. 2) We performed hydrogen-deuterium
exchange mass spectrometry (HDX-MS) analyses and observed HOX-dependent conformational
rearrangement in the $\alpha 3$ helix of CUT (**Page 7, Lines 236-243; Fig. 3f-i**).

2. *In light of highlighting the importance of the RR motif in this contribution, the characterization of*
*the OC2RR mutant is critical. Unfortunately, there is no explicit information on what kind of*
*mutation is behind the OC2RR variant. Please clarify this (I assume it's AA, as this was also used*
*for other mutants).*

Response:

We regret the oversight about not clearly describing the nature of mutations in the OC2RR mutant.
Both the arginines in the RR motif were indeed mutated to alanines. We have revised the text to make
this point clearer (**Page 8; Line 267-268**). In the revised manuscript, we have also mutated these
arginines individually and tested their DNA binding (please refer to our response to Reviewer 1 -
comment #5).

3. *The structural data of the OC2RR reveal a structure literally identical to the wt structure*
*(Extended Figure 4). The ITC data of the mutant (Figure 4) show an approximately 5-fold decrease*
*of binding affinity, similar to other mutants analyzed (OC2N, OC2SQ). Similarly, the kinetic data*

*show an increase of the DNA dissociation rate (Figure 5). Unfortunately, both the thermodynamic*
*and kinetic data are presented without repeated experiments and significance analysis. Therefore,*
*judging the significance at present is a matter of guessing, and proper addition of such analysis is*
*requested.*

Response:

We have included data in triplicates for both ITC and BLI-based binding experiments. We have
accordingly modified the respective tables to include the standard deviation representing these data.
The statistical significances, where applicable, have been indicated in **Extended Data Fig. 3a and**
**Extended Data Fig. 5b,c**. The repeat experiments are consistent with our original data and
conclusions drawn.

4. *Judging to what extent the data shown in Figure 5f-h is somewhat out of my own core expertise.*
*Nevertheless, I am wondering whether comparing endogenous (vector control) versus different*
*ONECUT2 version over expressed is still state-of-the-art (Figure 5f). Why was it not possible to*
*carry such experiments by complementation using a KO version as background?*

Response:

This is a valid question, and we appreciate this concern. In our earlier study (Rotinen et al.; [doi:](https://doi.org/10.1038/s41591-018-0241-1)
[10.1038/s41591-018-0241-1](https://doi.org/10.1038/s41591-018-0241-1)), we showed that OC2 is widely expressed in prostate cancer cells and
driver of prostate cancer cell growth. Knockout or knockdown of this gene significantly impairs the
cell growth. Notably, the LNCaP cell line (that we used here) has a relatively low OC2 expression
level, making it a dependable model to test OC2's function, and one that we and others in the field
have used in prior studies^{5,6}.

5. *PCR of KLK3 shows a much stronger effect for the OC2N and OC2RR mutants than for the*
*OC2SQ mutant (Figure 5g). This effect is not discussed in the text. Overall, as the last paragraph*
*of the results section addresses potential functional effects resulting from the mutations carried*
*out, I would have wished to see a more in-depth functional characterization, followed by*
*mechanistic explanations.*

Response:

We agree that in the case of *KLK3*, overexpression of OC2N and OC2RR show a stronger effect than
OC2SQ and the vector control (dominant negative effect). While the exact reason for this observation
is not clear, we have recently reported that OC2 acts as a chromatin remodeler and overexpression of
OC2 leads to a loss of promoter-enhancer contacts at *KLK3* locus (currently under peer-review and
available in BioRxiv⁷; [doi: 10.1101/2023.09.28.560025](https://doi.org/10.1101/2023.09.28.560025); refer to **Reviewer Fig. 2**, based on this
manuscript). CUT- and HOX-specific interactions with the DNA may have differential effects on
OC2-mediated chromatin remodeling. Thus, the effect of mutations in the CUT (OC2SQ) or HOX
(OC2N and OC2RR) on transcription at the chromatin remodeling sites may vary. Additionally, the
differences might also depend on loci-specific interactions of OC2 with other factors. Nevertheless, the
OC2 mutants we have tested do not show the effects associated with overexpression of the wild-type
OC2, thus implying the respective interactions to be relevant physiologically as well as in disease
progression. We have edited the text to reflect the above points (**Page 10; Lines 368-375**). Overall, the
purpose and design of our *in vitro* cell-based assays was more to assess the physiological relevance of
the interactions we have characterized than in terms of mechanism. However, as noted above, the

results might suggest a differential effect for CUT and HOX domains towards OC2-mediated
chromatin remodeling, which we intend to analyze further in the future.

[REDACTED]

Reviewer Fig. 2: HiChIP showing promoter-enhancer contacts at KLK3 gene locus. Promoter-enhancer contacts in cells without (Vec_con) and with OC2 overexpression (OC2_OE) in LNCaP cells are shown.

Please see Figure 4C in Qian, C. et al. ONECUT2 acts as a lineage plasticity driver in adenocarcinoma a
241 s well as neuroendocrine variants of prostate cancer. Nucleic Acids Res 52, 7740-7760 (2024). (DOI: 10.
1093/nar/gkae547.)

Additional specific comments:

6. *In the abstract it is stated that “evolutionarily conserved base interactions additionally determine
the thermodynamic stability of the ONECUT2-DNA complex”. I guess what the author are
referring to is “strength of DNA binding” or similar. “Thermodynamic stability” implies increased
temperature-dependent stability, which was not measured in this contribution.*

Response:

We have changed the phrase to “...ONECUT2-DNA binding energetics” (**Page 2; Lines 27-28**).

7. *Introduction: “OC2 is involved in other cancer types, including breast cancer, where it similarly
acts to drive lineage plasticity (manuscript in review).” Please provide this manuscript as
additional source for review or publish it in an accessible way (Biorxiv).*

Response:

This manuscript has just been accepted for publication in the journal Cellular Oncology and will be
available online shortly. We have attached a copy of this manuscript for the reviewer’s reference
(Reviewer attachment 1).

8. *Figure 2*

*Panel j: I guess the alignment has directly been taken from Jalview. I would have preferred an
alignment, which focuses more on key features discussed in this paper. Please explain the different
color codes, or refer to a reference where these color codes are defined. It is of note that the RR
motif does not belong to the most conserved residues within the alignment.*

Response:

We have modified the figure and highlighted only the residues analyzed in this manuscript.

Please refer to our response to Reviewer 1 – comment #4 about the RR motif conservation.

*Figures 3 and 4: The original ITC data could be moved into the Supplement. Proper statistical
and significance analysis is missing. The histogram presentation and inserted table show the same
content. Move one of the two into Supplement. It would be straight forward to merge the two
figures.*

Response:
The signature histograms have been moved to the Supplementary section as suggested. In Figure 3, the
revised manuscript has new panels from heat capacity and HDX-MS experiments. So, we thought it
might be appropriate to leave Figure 4 as it was. However, if the reviewer suggests otherwise, then
Figure 4 could be still merged into Figure 3.

*9. Figure 5: Proper statistical and significance analysis for data shown in panels a-d is missing.*

Response:
The standard deviations and significance analyses have been added.

*10. Extended Figure 1: Quantification of structural differences would be useful.*

Response:
We have added a table with the distances between phosphate backbones for the OC2-bound DNA and
a canonical B-DNA (**Extended Data Fig. 1b**).

*11. Extended Figure 3: This figure can be dropped as it only shows negative data.*

Response:
We have removed this figure.

*12. Supplementary Table 1: Please provide values with realistic precision (unit cell, Wilson B factor, R*
*merge, R meas etc.)*

Response:
We have modified the values as suggested.

**References:**

- 1. Lannoy, V.J., Burglin, T.R., Rousseau, G.G. & Lemaigre, F.P. Isoforms of hepatocyte nuclear
factor-6 differ in DNA-binding properties, contain a bifunctional homeodomain, and define the
new ONECUT class of homeodomain proteins. *J Biol Chem* **273**, 13552-62 (1998).
- 2. Gehring, W.J. et al. Homeodomain-DNA recognition. *Cell* **78**, 211-23 (1994).
- 3. Spolar, R.S. & Record, M.T., Jr. Coupling of local folding to site-specific binding of proteins to
DNA. *Science* **263**, 777-84 (1994).
- 4. Ha, J.H., Spolar, R.S. & Record, M.T., Jr. Role of the hydrophobic effect in stability of site-
specific protein-DNA complexes. *J Mol Biol* **209**, 801-16 (1989).
- 5. Guo, H. et al. ONECUT2 is a driver of neuroendocrine prostate cancer. *Nat Commun* **10**, 278
(2019).
- 6. Rotinen, M. et al. ONECUT2 is a targetable master regulator of lethal prostate cancer that
suppresses the androgen axis. *Nat Med* **24**, 1887-1898 (2018).
- 7. Qian, C. et al. ONECUT2 Activates Diverse Resistance Drivers of Androgen Receptor-
Independent Heterogeneity in Prostate Cancer. *bioRxiv* (2023).

VIEWER COMMENTS

Reviewer #1 (Remarks to the Author):

My previous comments have been adequately addressed.

Reviewer #2 (Remarks to the Author):

Overall comments:

When comparing the original manuscript version and the revised version it has become evident that various additional modifications have been introduced that are not highlighted. The referee requests a complete overview of changes been made, text and figures, including the Extended Data Figures. It would also be helpful, to receive a comparison table, indicating which Figures (Extended Data Figures included) of the revised version of the manuscript correspond to those of the original version. In the absence of this, it is literally impossible to review all changes included.

In my view, subject for further editorial review, there is a general misconception on hybrid figures and tables. Just focusing on the figures of the main part only, Fig. 3d+e, Fig. 4d, Fig. 5e, should be presented as separate tables. All quantitative data should contain significance statistics. When not possible, the reasons need to be explained. Decimal values should be presented with realistic precision. Values should be presented consistently in all tables, "e-*" expressions should be avoided. There are straight forward solutions to this, such as expressing KD's e.g. in nM instead of M (example of Fig. 3d). Several subscripts/superscripts are missing. As an example, in Fig. 3d, for the dissociation constant KD, the "D" should be expressed as subscript, etc. In some tables, dimensions are missing, such Extended Data Fig. 1b, I guess the dimension would be [Å]. Plots should be shown with identical scales (for example, Fig. 3a) when presented next to each other, implying direct comparison. My impression is that several figure components, beyond the structural figures, are directly copied from external graphics and visualization packages, as commented before. Those packages need to be named. In some of them, no care has been taken to ensure a consistent nomenclature, e.g. "OC1" is "HNF6" in Fig. 2j, etc. Overall I have been surprised to find so many formatting and text errors/typos when reviewing a revised version of a paper for a peer reviewed journal. I do request from the authors to be more thorough by internal review.

In the following, there are comments to replies by authors on previous comments of this reviewer.

1) I did not ask to add a heat capacity analysis. The data are interesting but not unexpected. DNA binding of a single domain (CUT) obviously leads to less folding events than DNA binding of OC2, which includes two domains with a flexible linker in between. I also did not request any additional HDX-MS experiments but they are appreciated as well. Unfortunately, the way they are presented in Fig. 3f-i is quite in-comprehensive. For instance, panel h shows a black/white color scale presentation but the scale bar below is for a red/blue range. I am not able to extract clear-cut take home messages from this figure. My request, however, was to analyze the structural differences along the matching sequences and interpret these differences taking estimated positional experimental errors from the methods used to determine the respective structures into account, to allow an assessment of their significance. This has not been done.

3) Have repeated experiments been biological (e.i. using different sample preps) or technical replicates? I don't understand the statement "The repeat experiments are consistent with our original data and conclusions drawn." How can the conclusion in terms of significance be the same, when there was no basis to test significance (original version)?

7) "(accepted for publication)" should be replaced by a proper reference. This is one of the many changes in the manuscript, which was not highlighted (see comment above), still including a statement "accepted for publication" that (to the best of my knowledge) is not permissible. I am somewhat surprised about such reference both in the original and revised versions, given that this manuscript was corresponded by a different group. Did the authors have permission for this?

8) The overall impression for a lack of a thoughtful composition of the main findings in the main figures remains (see comments above).

12) Even if this has been corrected, various other estimates are still unrealistic (see comments above).

We thank the reviewers for their time and comments on our revised manuscript. We have provided a point-wise response to the reviewers' comments in the following sections.

In the revised manuscript, the changes are shown in two colors, Green and Blue. The changes made in the previous submission are maintained in Green. Any other changes including those that were not highlighted before are shown in Blue.

Reviewer #1 (Remarks to the Author):

My previous comments have been adequately addressed.

Authors' response: We thank the reviewer, and we are pleased that we could address all the comments.

Reviewer #2 (Remarks to the Author):

Overall comments:

1. *When comparing the original manuscript version and the revised version it has become evident that various additional modifications have been introduced that are not highlighted. The referee requests a complete overview of changes been made, text and figures, including the Extended Data Figures.*

Authors' response: We thank the reviewer for the detailed critiques. We have revised the manuscript carefully and addressed all the points raised. As requested, below is an overview of all the changes compared to the original version (the line nos. in the table below are consistent with the current manuscript version).

Sections	1st Revision	2nd Revision (current)
Abstract	1. Verbiage change was made (line 27)	1. Additional verbiage changes have been made (lines 18, 19, 21, 24-25, 26)
Introduction	1. Verbiage changes were made (lines 58-59, 60, 64, 68, 69)	1. Reference for OC2 in ovarian cancer (Reference #29) is included (line 59)
Results	Main Figures 1. All measurements reported in the tables were presented with statistics (according to experiments performed in triplicates) 2. Alignments were modified to highlight only the residues studied in this manuscript 3. ITC signature plots were moved to supplementary section	1. All tables previously in the main figures have been presented separately (Tables 1-4)

-
4. Figure panels based on heat capacity and HDX-MS experiments were added
 5. Kinetics experiments were re-done with slightly altered concentration range (proteins titrated at 75-275 nM instead of 50-250 nM) than original manuscript
 6. Kinetics data were fitted with 1:1 binding model
-

Results Text

- | | |
|--|---|
| 1. More detailed analysis of the RR motif was added (lines 176-184)2. Thermodynamics and kinetics parameters were edited according to the results from triplicate experiments3. Text related to heat capacity (lines 226-238) and HDX experiments were added (lines 240-248)4. Text describing ITC (lines 299-306) and kinetics assays (lines 339-354) for OC2R479A & OC2R480A single mutants were added5. Discussion on stronger effects of OC2 mutations on AR targets was added (lines 375-382) | 1. Positional errors to evaluate differences between OC2 NMR and crystal structures is included (lines 102-105)2. Figure numbers have been edited in text to reflect changes in Figure compositions3. Verbiage edits have been made including appropriate subscripts, as suggested |
|--|---|
-

Supplementary

Supplementary Figures

- | | |
|--|--|
| 1. Quantification of differences in bound and unbound DNA was presented (Extended Data Fig. 1b)2. Signature plots were moved to supplementary figures (Extended Data Fig. 3a & 5c)3. Heat capacity graph was added (Extended Data Fig. 3b)4. HDX-MS coverage maps were included (Extended Data Fig. 3c,d)5. Electron densities for the key residues were included (Extended Data Fig. 4b)6. Thermodynamic parameters and comparison for R479A and R480A single mutants were added (Extended Data Fig. 5a,b)7. Kinetics parameters of R479A and R480A single mutants were added (Extended Data Fig. 7a-c, h-i) | 1. Unit (Å) for the differences in bound and unbound DNA is shown (Extended Data Fig. 1b) |
|--|--|
-

	8. Lack of binding of HOX domain alone to the DNA verified using kinetics (Extended Data Fig. 7j)
	Supplementary Tables 1. New table for HDX-MS analysis was added (Supplementary Table 2)
Discussion	1. Verbiage changes were made (lines 412 & 418) 2. Last line of paragraph 2 in original version was removed (line 422) .
Methods	1. HDX-MS method was added (lines 908-938) 2. New Kinetics experiments were done with an additional component (0.005% Tween-20) in the buffer (line 901) 1. Notes on nature of replicates, significance analysis and softwares used for preparing figure panels are added/highlighted 2. HDX-MS method is edited for correct subscripts

3. *It would also be helpful, to receive a comparison table, indicating which Figures (Extended Data Figures included) of the revised version of the manuscript correspond to those of the original version. In the absence of this, it is literally impossible to review all changes included.*

Authors' response: Below are tables comparing the original and new figures as requested:

Main Figures^s/Tables

Figure no.	Panel nos. (original)/description	Comments	Current version
Figure 1	-		Unchanged
Figure 2	j sequence alignments	The whole alignments were shown in Clustal color scheme	- Only important residues have been highlighted in Clustal scheme - 'HNF6' changed to 'OC1'
Figure 3	b-c ITC isotherms	Were in the same row as panel a	Moved below panel a
	d ITC signature plots	Moved	Moved to Extended Data Fig. 3a [†]
	e ITC table	Moved	Moved to Table 1 ^{††}

		New panels added	Panels d-g ; - new panels added showing the HDX-MS data - panel g key edited from “CUT 40:1 DNA” to “CUT + DNA” for consistency
Figure 4	d ITC signature plots	Moved	Moved to Extended Data Fig. 5c ^{†††}
	e ITC table	Moved	Moved to Table 3 ^{††}
Figure 5	e Kinetics table	Moved	Moved to Table 4 ^{††}
	f-h Cell-based assay data		Are now panels e-g
Tables 1, 3 and 4			- Were part of Figures 3, 4 & 5 (noted above ^{††}) - Table 4: k_d and k_a units are now written as ($M^{-1}s^{-1}$) and (s^{-1}) respectively [previously ($1/M*s$) and ($1/s$)], for consistency with the results text
Table 2			New table showing heat capacity data

Extended Data Figures[§]/Tables

Supplemental Fig no.	Panel nos. (original)/ description	Comments	Current version
Extended Data Fig 1	a Alignment of OC2 bound and unbound DNA	CUT and HOX domains were not labeled	Labels added
		New panel added	Panel b ; New panel showing quantification of the differences
	b-e structural alignments & PEG10 sequence		are now panels c-f
Extended Data Fig 2	-		Unchanged

Extended Data Fig 3	a-b negative ITC data	Removed	New Extended Data Fig 3 shows panels (a) ITC signature plots (noted above [†]), (b) heat capacity graph and (c-d) HDX-MS peptide coverage maps
Extended Data Fig 4	a sequence alignments	The whole alignments were shown in Clustal color scheme	 - Only important residues have been highlighted in Clustal scheme - Antp HOX domain added to the HOX alignment (bottom) - An underscore (“_”) included in the HOX protein identifiers (bottom) for consistency - MATα2_HOX/41-99 replaced with MATα2_HOX/41-99
	b	Moved	 - Moved to Extended data Fig. 6[#] - New Panel b now shows the electron densities of the residues investigated in this manuscript
Extended Data Fig 5	a-d kinetics curve fittings and mRNA expression levels	Moved	 - Moved to Extended data Fig. 7^{##} - New figure shows (a) thermodynamics table for OC2R479A and OC2R480A mutants, (b) K_D comparison graph from ITC and (c) signature plots of mutants (noted above^{†††})
Extended Data Fig 6	a-b protein purification profiles	Moved	 - Moved to Extended Data Fig 8^{###} - Was part of Extended data Fig 4b (noted above[#])
Extended Data Fig 7			Panels a-m; New panels showing (a-c) kinetics data of OC2R479A and OC2R480A mutants, (d-i) graphs showing kinetics curve fittings (noted above^{##}), (j) HOX-DNA binding kinetics, (k-l) graphs showing k_a and k_d comparisons of OC2 and its mutants and (m) relative mRNA expression levels of OC2 and the mutants
Extended Data Fig 8			Moved from Extended Data Fig. 6 (noted above ^{###})

§ The figure legends have been edited according to these changes

4. *In my view, subject for further editorial review, there is a general misconception on hybrid figures and tables. Just focusing on the figures of the main part only, Fig. 3d+e, Fig. 4d, Fig. 5e, should be presented as separate tables. All quantitative data should contain significance statistics. When not possible, the reasons need to be explained.*

Authors' response: Based on this suggestion, the tables in the main figures have now been presented separately (**Tables 1-4; pages 24-27**). We have provided statistics/significance for all data in the tables and graphs.

5. *Decimal values should be presented with realistic precision. Values should be presented consistently in all tables, "e-*" expressions should be avoided. There are straight forward solutions to this, such as expressing KD's e.g. in nM instead of M (example of Fig. 3d).*

Authors' response: We have made the necessary changes as below:

- a) All K_D values are now presented in nM and approximated to the nearest whole numbers (**Tables 1 & 3, Extended Data Fig. 5a**)
- b) All ΔH , ΔG and $-T\Delta S$ values are approximated to one decimal point (**Tables 1-3; Extended Data Fig. 5a**)
- c) Kinetics parameters - k_a are presented as whole numbers (**Table 4; Extended Data Fig. 7c**)
- k_d are presented up to four decimal places (this was necessary in order to avoid logarithmic expressions as suggested while enabling comparison, since the dissociation rates are extremely slow, especially for the wild-type OC2) (**Table 4; Extended Data Fig. 7c**)

6. *Several subscripts/superscripts are missing. As an example, in Fig. 3d, for the dissociation constant K_D , the "D" should be expressed as subscript, etc. In some tables, dimensions are missing, such Extended Data Fig. 1b, I guess the dimension would be [Å].*

Authors' response: Based on this suggestion, following changes have been incorporated

- a) 'KD' to ' K_D ', 'ka' to ' k_a ' and 'kd' to ' k_d '
- b) edited HDX-MS method (**lines 908-938**) for proper subscripts
- c) added the appropriate unit (Å) in **Extended Data Fig. 1b**

7. *Plots should be shown with identical scales (for example, Fig. 3a) when presented next to each other, implying direct comparison.*

Authors' response: In **Fig. 3a-c**, the DP ($\mu\text{cal/s}$) plots (top) could not be scaled identically as explained here. In these panels (Fig. 3a-c) the DNA binding of OC2, CUT, and CUT+HOX are compared. The ΔH (kcal/mol; bottom plots) is the predominant measure to be compared between protein variants, and since ΔH is normalized to protein concentration these y-axes are identically scaled between graphs. The DP ($\mu\text{cal/s}$; top plots) is a measure of raw heat generated by the interaction, should be greater than [0.1] during early injections to provide adequate signal-to-noise, and is a value not normalized to sample concentrations. Since OC2 binding to DNA generates a large amount of heat, only 10 μM OC2 protein concentration is needed to meet this signal-to-noise

threshold, while the isolated subdomain CUT generates less heat and requires a higher concentration (50 μM) to meet the DP signal-to-noise threshold. At the same time, 10 μM OC2 is the maximum concentration that could be used with OC2 since a higher protein concentration would lead to a C value ($=[\text{protein}]/K_D$) $\gg 1000$ and would provide an unreliable K_D . Since DP is relative to protein concentration, the DP ($\mu\text{cal/s}$) are not able to have the same scales when different protein concentrations are used (though in Fig. 3, CUT and CUT+HOX have the same DP scale since the same protein concentration is used in these experiments).

Since panels 3a-c cannot be scaled identically, we have now moved **Panels 3b and 3c below Panel 3a**, to avoid confusion for the readers.

We have checked all other graphs & plots in the manuscript to ensure they have been presented with identical scales.

8. *My impression is that several figure components, beyond the structural figures, are directly copied from external graphics and visualization packages, as commented before. Those packages need to be named.*

Authors' response: Below is a list of panels that were exported from the softwares. These have now been noted appropriately in the methods section as mentioned below.

- a) Sequence alignments – Jalview (noted in methods; **lines 941-942**)
- b) ITC raw heats and isotherms – MicroCal PEAQ-ITC Analysis software (noted in Methods; **lines 890-892**)
- c) HDX-MS uptake plot (Fig. 3g) – DECA (noted in methods; **lines 937-938**)
- d) HDX-MS peptide coverage (Extended Data Fig. 3 c,d) – DynamX (noted in methods; **lines 931-932**)

All remaining graphs/plots were prepared in GraphPad Prism and these have been noted accordingly in the relevant methods sections.

9. *In some of them, no care has been taken to ensure a consistent nomenclature, e.g. “OC1” is “HNF6” in Fig. 2j, etc. Overall I have been surprised to find so many formatting and text errors/typos when reviewing a revised version of a paper for a peer reviewed journal. I do request from the authors to be more thorough by internal review.*

Authors' response: We have changed HNF6 to OC1 in Figure 2j. We have carefully gone through and revised the manuscript to correct any such inconsistencies as noted in our above responses.

In the following, there are comments to replies by authors on previous comments of this reviewer.

10. *I did not ask to add a heat capacity analysis. The data are interesting but not unexpected. DNA binding of a single domain (CUT) obviously leads to less folding events than DNA binding of OC2, which includes two domains with a flexible linker in between. I also did not request any additional HDX-MS experiments but they are appreciated as well. Unfortunately, the way they are presented in Fig. 3f-i is quite in-comprehensive. For instance, panel h shows a black/white color scale presentation but the scale bar below is for a red/blue range. I am not able to extract clear-cut take home messages from this figure.*

Authors' response: We appreciate this concern and take this opportunity to clarify the figure further (note that in the current manuscript version, the HDX-MS figure panels are **Fig. 3d-g**). We would like to point out that the color keys for both panels, showing either OC2 (Fig. 3e) or CUT (Fig. 3f), are the same, as explained below. The color scheme Blue-White-Red depicts decreased to increased deuterium uptake of DNA-bound relative to unbound protein ('White' showing neutral or no change in uptake) while 'Black' represents regions of the protein that did not show coverage in the HDX-MS experiment (described in the respective figure legend; **lines 674-676**). The isolated CUT domain, unlike intact OC2, shows no DNA-dependent change in deuterium uptake. Consequently, the intact OC2 and CUT are colored according to this difference in their uptake patterns (CUT being neutral while OC2 showing decreased uptake in presence of the DNA).

To avoid confusion, however, we have now removed the color key from both the panels 3e and 3f; explanation for the color rendering is provided in the respective figure legend. In addition, to make above points clearer to the reader, we have included in the figure legend the following sentence (**lines 680-682**):

“There were no significant differences in deuterium uptake upon DNA binding of the CUT domain alone (consequently, mostly white except the black regions that showed no coverage)”

11. My request, however, was to analyze the structural differences along the matching sequences and interpret these differences taking estimated positional experimental errors from the methods used to determine the respective structures into account, to allow an assessment of their significance. This has not been done.

Authors' response: We regret this oversight and have now added a root mean square deviation (rmsd)-based analysis. Structural alignment of the respective apo- and DNA-bound CUT domains showed a root mean square deviation (rmsd) of 3.3 Å, compared to the average rmsd of 1.5 - 2.5 Å observed between the structures of the same proteins elucidated by NMR and X-ray crystallography, as reported by Sikic et al. (*Open Biochem J* **4**, 83-95 (2010); Reference #31). These lines have been added to the manuscript text (**lines 102-105**).

12. Have repeated experiments been biological (e.i. using different sample preps) or technical replicates? I don't understand the statement "The repeat experiments are consistent with our original data and conclusions drawn." How can the conclusion in terms of significance be the same, when there was no basis to test significance (original version)?

Authors' response: We have now clarified in the manuscript methods and figure legends that the biophysical experiments (ITC, kinetics and HDX-MS) were performed in technical replicates while the cell-based assays were performed with biological replicates.

13. "(accepted for publication)" should be replaced by a proper reference. This is one of the many changes in the manuscript, which was not highlighted (see comment above), still including a statement "accepted for publication" that (to the best of my knowledge) is not permissive. I am somewhat surprised about such reference both in the original and revised versions, given that this manuscript was corresponded by a different group. Did the authors have permission for this?

Authors' response: We regret not highlighting this change. The requested reference has now been provided in the manuscript text (Reference #29; **line 59**).

14. The overall impression for a lack of a thoughtful composition of the main findings in the main figures remains (see comments above).

15. Even if this has been corrected, various other estimates are still unrealistic (see comments above)

Authors' response: We appreciate the thoughtful feedback on the figures by the Reviewer. Based on the suggestions, we have now edited the figures as noted in our responses above. We have also made the necessary changes regarding the realistic precision of values presented (noted in our response to comment #4 above). We believe the changes listed in our responses above have addressed the concerns appropriately.

REVIEWERS' COMMENTS

Reviewer #2 (Remarks to the Author):

In this second revision, all comments have been reasonably addressed.

@ Comment 11: overall structural statistics are now given in this version. However, my previous request (although not explicitly stated) was to discuss specific structural changes presented in this manuscript in light of overall statistics. As it stands by now, the stats presented are nice to have but without meaning on specific content of the manuscript.

Reviewer #2 (Remarks to the Author):

In this second revision, all comments have been reasonably addressed.

Authors' response: We thank the reviewer and are pleased to have been able to address all the comments.

@ Comment 11: overall structural statistics are now given in this version. However, my previous request (although not explicitly stated) was to discuss specific structural changes presented in this manuscript in light of overall statistics. As it stands by now, the stats presented are nice to have but without meaning on specific content of the manuscript.

Authors' response: Based on this comment, we have now edited the text as follows (the changes have been shown in **Blue** in manuscript text):

“Structural alignment of the respective apo- and DNA-bound CUT domains showed a root mean square deviation (rmsd) of 3.3 Å. This value exceeds the average rmsd of 1.5 - 2.5 Å observed between the structures of the same proteins elucidated by NMR and X-ray crystallography³², suggesting the structural differences between the apo- and DNA-bound forms are induced by the DNA and independent of the respective techniques.”